# Enhancing intracellular accumulation and target engagement of PROTACs with reversible covalent chemistry

Wen-Hao Guo[1,13], Xiaoli Qi[1,13], Xin Yu[1], Yang Liu[2], Chan-I Chung[3], Fang Bai[4], Xingcheng Lin[5], Dong Lu[1], Lingfei Wang[1], Jianwei Chen[1], Lynn Hsiao Su[1], Krystle J. Nomie[2], Feng Li [6,7], Meng C. Wang [8,9,10], Xiaokun Shu [3], José N. Onuchic[4], Jennifer A. Woyach[11], Michael L. Wang [2] & Jin Wang [1,7,12✉]

Current efforts in the proteolysis targeting chimera (PROTAC) field mostly focus on choosing an appropriate E3 ligase for the target protein, improving the binding affinities towards the target protein and the E3 ligase, and optimizing the PROTAC linker. However, due to the large molecular weights of PROTACs, their cellular uptake remains an issue. Through comparing how different warhead chemistry, reversible noncovalent (RNC), reversible covalent (RC), and irreversible covalent (IRC) binders, affects the degradation of Bruton's Tyrosine Kinase (BTK), we serendipitously discover that cyano-acrylamide-based reversible covalent chemistry can significantly enhance the intracellular accumulation and target engagement of PROTACs and develop RC-1 as a reversible covalent BTK PROTAC with a high target occupancy as its corresponding kinase inhibitor and effectiveness as a dual functional inhibitor and degrader, a different mechanism-of-action for PROTACs. Importantly, this reversible covalent strategy is generalizable to improve other PROTACs, opening a path to enhance PROTAC efficacy.

[1] Department of Pharmacology and Chemical Biology, Baylor College of Medicine, Houston, TX 77030, USA. [2] Division of Cancer Medicine, Department of Lymphoma/Myeloma, The University of Texas MD Anderson Cancer Center, Houston, TX 77030, USA. [3] Department of Pharmaceutical Chemistry, University of California—San Francisco, San Francisco, CA 94158, USA. [4] Center for Theoretical Biological Physics, Rice University, Houston, TX 77005, USA. [5] Department of Chemistry, Massachusetts Institute of Technology, Cambridge, MA 02139, USA. [6] Department of Pathology and Immunology, Baylor College of Medicine, Houston, TX 77030, USA. [7] Center for Drug Discovery, Baylor College of Medicine, Houston, TX 77030, USA. [8] Department of Molecular and Human Genetics, Baylor College of Medicine, Houston, TX 77030, USA. [9] Huffington Center on Aging, Baylor College of Medicine, Houston, TX 77030, USA. [10] Howard Hughes Medical Institute, Baylor College of Medicine, Houston, TX 77030, USA. [11] Division of Hematology, Department of Internal Medicine, The Ohio State University, Columbus, OH 43210, USA. [12] Department of Molecular and Cellular Biology, Baylor College of Medicine, Houston, TX 77030, USA. [13] These authors contributed equally: Wen-Hao Guo, Xiaoli Qi. ✉email: wangj@bcm.edu

A proteolysis targeting chimera (PROTAC) is a hetero-bifunctional molecule that can bind both a target protein and an E3 ubiquitin ligase to facilitate the formation of a ternary complex, leading to ubiquitination and ultimate degradation of the target protein[1–5]. Compared with oligonucleotide and CRISPR therapeutics that face in vivo delivery challenges, PROTACs are small molecule therapeutics that provide opportunities to achieve broadly applicable body-wide protein knockdown. Protein degraders have many advantages compared with traditional small molecule inhibitors, including sub-stoichiometric target occupancy[6,7], sustainable pharmacodynamic effects even without constant PROTAC exposure[8], degradation of the full-length protein to reduce the possibility to develop drug resistance through mutations or compensatory protein overexpression and accumulation[9,10], and enhanced specificity controlled by protein–protein interactions between the targeted protein and the recruited E3 ligase[11–14].

Most PROTACs reported thus far have been based on non-covalent binding to their target proteins. Although irreversible covalent inhibitors, such as ibrutinib, have achieved tremendous clinical success based on their strong target affinities and high target occupancies, it was recently reported that PROTACs with irreversible covalent binders to targeted proteins failed to induce efficient protein degradation[15]. Although the potential mechanism accounting for the inhibition of protein degradation was not elucidated, it was postulated that irreversible covalent PROTACs are unable to induce protein degradation in a sub-stoichiometric/catalytic manner because they are consumed once they bind to their targeted protein. However, there are also examples arguing against this hypothesis[7,16–19].

Reversible covalent chemistry previously enabled our development of a series of fluorescent probes that can quantify glutathione concentrations in living cells[20–24]. Here, we assess whether reversible covalent PROTACs can not only enhance the binding affinity to targeted protein but also overcome the one-shot deal drawback of irreversible covalent PROTACs (Fig. 1). Beginning as a basic science exploration to compare how the warhead chemistry of PROTACs with reversible noncovalent (RNC), reversible covalent (RC), and irreversible covalent (IRC) binders affect protein degradation, we choose Bruton's tyrosine kinase (BTK) as a model target to test the hypothesis. To our

surprise, we discover that cyano-acrylamide-based reversible covalent binder to BTK can significantly enhance drug accumulation and target engagement (TE) in cells. Building on this discovery, we develop RC-1 as a reversible covalent BTK PROTAC with a high target occupancy as its corresponding kinase inhibitor. RC-1 is effective as a dual functional inhibitor and degrader, providing a different mechanism-of-action for PROTACs, and forms a stable ternary complex by reducing protein conformational flexibility compared with the noncovalent PRTOAC counterparts. Importantly, we find that this reversible covalent strategy can be generalized and applied to improve other PRO-TACs. We hope that this study adds another dimension to improve the cellular efficacy of PROTACs.

## Results

**Comparison of different warhead chemistry for BTK PROTACs.** As the first FDA-approved covalent kinase inhibitor, Ibrutinib irreversibly reacts with the free cysteine residue (C481) in the active site of BTK to form a covalent bond[25], but can still bind to the C481S BTK mutant mostly through hydrogen bonding with >40 folds lower affinity[26]. Johnson et al.[27] showed that ibrutinib is >6 folds more potent than its Michael acceptor saturated ibrutinib analog in a kinase inhibition assay for wild-type BTK ($IC_{50}$ 0.72 nM vs 4.9 nM), while both compounds are equally potent towards BTK C481S mutant ($IC_{50}$ 4.6 nM vs 4.7 nM). To compare how different warhead chemistry may affect the target protein degradation, we design three PROTAC molecules, RC-1, RNC-1, and IRC-1, which form reversible covalent (RC), reversible noncovalent (RNC), and irreversible covalent (IRC) binding to BTK (Fig. 2a). Following previous work[15,28–32], we choose pomalidomide as the CRBN E3 ligase binder. RC-1, RNC-1, and IRC-1 share the same linker and E3 ligand binder and differ only by the binding moieties to BTK. We choose MOLM-14, an acute myeloid leukemia cell line, as a model system to study PROTAC-mediated BTK degradation following previous work[29].

To test whether the three PROTACs can induce BTK degradation, we treated MOLM-14 cells with RC-1, RNC-1, and IRC-1 for 24 h, followed by western blot to quantify the total BTK levels (Fig. 2b). Consistent with Tinworth et al.'s[15] recent report, IRC-1 induced inefficient BTK degradation. Comparing RC-1 and RNC-1, RC-1 was more potent for BTK degradation than RNC-1 at lower concentrations (8 and 40 nM) and comparable to RNC-1 at 200 nM. RC-1 potently induced BTK degradation in MOLM-14 cells ($DC_{50}$ = 6.6 nM, Fig. 2c), as one of the most potent BTK degraders reported so far[15,30]. Additionally, neither the RC-1 warhead, nor pomalidomide, nor a combination of both caused BTK degradation (Fig. 2d), indicating that the bifunctional PROTAC molecule is essential to facilitate the formation of a ternary complex of {BTK-PROTAC-CRBN} to induce BTK degradation.

**Linker optimization for reversible covalent BTK PROTAC.** Encouraged by the promising result for RC-1, we set out to optimize the linker between the BTK reversible covalent binder and pomalidomide (Fig. 2e). Recently, Zorba et al.[30] showed that increasing PROTAC linker length alleviates steric clashes between BTK and CRBN and improves the efficacy of BTK degradation, indicating the absence of thermodynamic cooperativity in the formation of a ternary complex of {BTK-PROTAC-CRBN}. Therefore, we synthesized RC-2, RC-3, RC-4, and RC-5, which are 1, 2, 5, and 8 atoms longer in the linker length compared with RC-1. Contrary to Zorba et al.'s[30] finding, we found that RC-1 is the most efficacious for BTK degradation compared with the PROTACs possessing longer linkers, suggesting cooperative ternary complex formation (Supplementary Fig. 1).

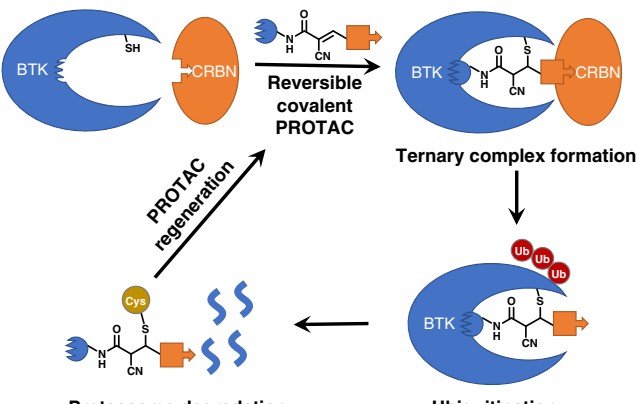

**Fig. 1 Demonstration of catalytic degradation of targeted proteins by reversible covalent PROTACs.** The premise of this reversible covalent PROTAC design is the weak reactivity (mM $K_d$) between α-cyano-acrylamide group (the chemical structure shown above) and free thiols. Only when the PROTAC molecule binds to the active site of the targeted protein, the nearby cysteine side chain can react with the α-cyano-acrylamide group to form a stable covalent bond. Once the targeted protein is degraded, the reversible covalent PROTAC can be regenerated.

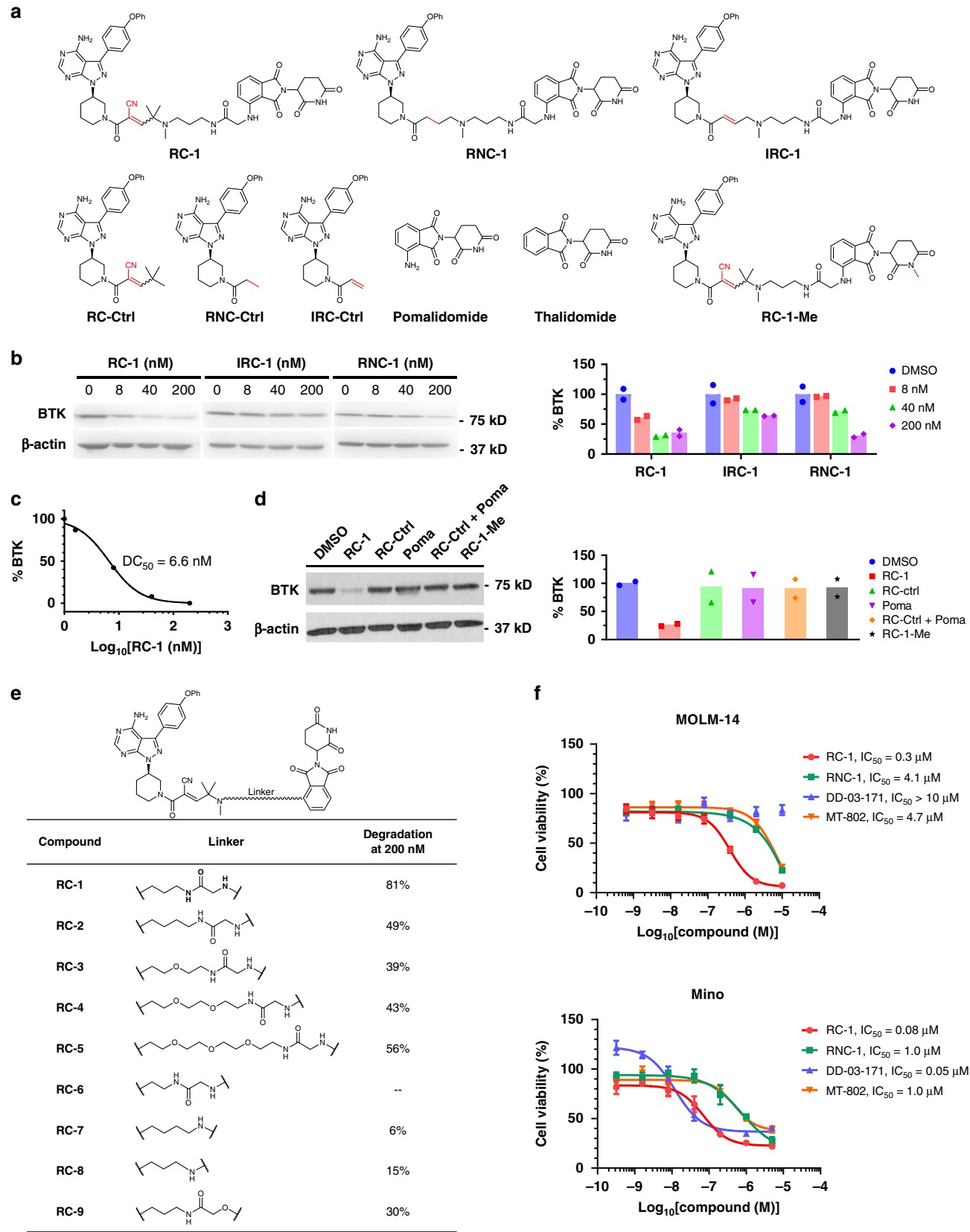

To test whether the BTK degradation efficacy of RC-1 can be improved through further reducing the linker length, we designed RC-6, RC-7, and RC-8, which are 1, 2, and 3 atoms shorter in the linker length compared with RC-1. Unfortunately, we could not synthesize RC-6 due to an intramolecular reaction between the amide and the Michael acceptor. Comparing the BTK degradation capability in MOLM-14 cells, we found that RC-7 and RC-8 are inferior to RC-1, possibly due to unfavorable steric clashes between BTK and CRBN. The efficacy of BTK degradation decreases significantly through only a single atom change of the

**Fig. 2 BTK degradation induced by PROTACs. a** Chemical structures of BTK degraders and their controls. RC-1, RNC-1, and IRC-1 are BTK degraders with reversible covalent, reversible noncovalent, and irreversible covalent warheads, respectively. RC-Ctrl, RNC-Ctrl, and IRC-Ctrl (i.e., ibrutinib) are the corresponding warhead controls for RC-1, RNC-1, and IRC-1, respectively. **b** MOLM-14 cells were incubated with RC-1, RNC-1, and IRC-1 at different concentrations (0 (blue), 8 (red), 40 (green), and 200 (purple) nM) for 24 h. The BTK levels were quantified by western blotting. Duplicates were performed. **c** RC-1 dose-dependent BTK degradation in MOLM-14 cells. $DC_{50}$: compound concentration inducing 50% of protein degradation. Duplicates were performed. **d** MOLM-14 cells were treated with DMSO (blue), RC-1 (light red), RC-Ctrl (light green), Pomalidomide (purple), RC-Ctrl + Pomalidomide (orange), and RC-1-Me (black) for 24 h. All the compound concentrations are 200 nM. Neither RC-Ctrl, nor pomalidomide, nor a combination of both caused BTK degradation, indicating that the bifunctional PROTAC molecule is essential to facilitate the formation of a ternary complex of {BTK-PROTAC-CRBN} in order to induce BTK degradation. Duplicates were performed. **e** Linker optimization for reversible covalent BTK degraders. Optimization of the linkers of the RC series of BTK degraders and their corresponding percentage of BTK degradation in MOLM-14 cells (200 nM, 24 h incubation). Note: RC-6 was unstable due to intramolecular cyclization with the Michael acceptor. **f** Comparison of RC-1 (red circle) and RNC-1 (green square) with other reported BTK degraders. Two previously reported BTK degraders DD-03-171 (blue pyramid) and MT-802 (orange inverted pyramid) were synthesized in house. The cells were treated with compounds for 72 h. Then Alarma blue assays were performed to quantify the cellular viabilities. Data are presented as mean values ± SEM ($n = 5$ biologically independent samples). Source data are provided as a Source Data file.

---

### Table 1 BTK inhibition, in-cell target engagement and cell viability.

| Compound | $K_d$ (nM)[a] | BTK inhibition $IC_{50}$ (nM)[b] | Cell viability $IC_{50}$ (µM)[c] | BTK TE $IC_{50,TE}$ (µM)[d] | $K'_{P,D}$[e] |
|---|---|---|---|---|---|
| RC-1 | 6.4 | 1.8 | 0.31 | 0.043 | $1.5 \times 10^{-1}$ |
| IRC-1 | 3.0 | 0.8 | 2.7 | 0.13 | $2.3 \times 10^{-2}$ |
| RNC-1 | 4.7 | 21.2 | 4.1 | 1.27 | $3.7 \times 10^{-3}$ |
| RC-1-Me | – | 6.8 | 0.21 | – | – |
| RC-Ctrl | 9.1 | 2.2 | 0.50 | 0.059 | $1.5 \times 10^{-1}$ |
| IRC-Ctrl | – | 0.3 | 0.33 | – | – |
| RNC-Ctrl | – | 13.6 | 0.42 | – | – |
| RNC-1-CN-DiMe | 29 | – | – | 0.93 | $3.1 \times 10^{-2}$ |
| IRC-1-DiMe | 11 | – | – | 0.38 | $2.9 \times 10^{-2}$ |

[a]The dissociation equilibrium constant $K_d$ was measured using the full-length BTK protein by Eurofins DiscoverX. The reported $K_d$ values for RC-1, IRC-1, and RC-Ctrl were measured in the absence of DTT in the buffer. DTT (6 mM) used in the standard assay condition increases the $K_d$ values of RC-1 and RC-Ctrl by 2–3 folds, while it does not affect the measurement for IRC-1. The reported $K_d$ value for RNC-1 was measured in the standard assay buffer in the presence of DTT. Duplicates were performed. For more information please see Supplementary Data 1.
[b]The biochemical BTK inhibition (BTK Inhibition $IC_{50}$) was measured using the BTK assay kit from AssayQuant Technologies Inc. Duplicates were performed.
[c]Cell viability $IC_{50}$ was performed by treating MOLM-14 cells with compounds for 72 h, followed by Alamar Blue assays.
[d]BTK Target Engagement $IC_{50}$ is the concentration of an unlabeled compound that results in a half-maximal inhibition binding of the BTK tracer. The target engagement of compounds was assessed following Promega's assay protocol. Triplicates were performed.
[e]$K'_{P,D}$ is a relative intracellular accumulation coefficient for drug D and calculated as $K_d/IC_{50,TE}$. $K'_{P,D}$ is an assay-dependent parameter to quantify the tendency of intracellular accumulation of a drug. Under the same assay conditions, a greater $K'_{P,D}$ value for a drug reflects its higher tendency to accumulate inside cells. Please refer to the Methods section for detailed explanation.

---

thalidomide aryl amine nitrogen to an oxygen (RC-9; Supplementary Fig. 1).

In Buhimschi et al.'s[31] recent study, a BTK PROTAC MT-802 has the linker placed at the C5 position on the phthalimide ring of pomalidomide instead of the C4 position as in RC-1. We synthesized RC-10 by placing the linker at the C5 position of the phthalimide ring and found that RC-10 cannot induce any BTK degradation (Supplementary Fig. 1). Therefore, we concluded that RC-1 has the optimal linker length and position for BTK degradation with the BTK and CRBN binders used.

**Quantification of BTK and CRBN concentrations in cells**. The Crews group developed a series of PROTACs for TANK-Binding Kinase 1 (TBK-1) with different binding affinities toward TBK-1 and von Hippel-Lindau (VHL)[33]. We re-analyzed the data in this study and found a strong correlation between binding affinities towards TBK-1 and the $DC_{50}$ values in cells (Supplementary Fig. 2), suggesting that tighter binding to the target protein leads to more efficient PROTAC-induced protein degradation in cells. In our study, it is reasonable to assume that the binding affinity of RC-1 to BTK is higher than RNC-1 due to the covalent bond formed between RC-1 and BTK. It could be taken for granted that tighter binding to BTK leads to more formation of the ternary complex, resulting in more efficient BTK degradation. However, the formation of the {BTK-PROTAC-CRBN} ternary complex depends on not only the binding affinities of PROTAC toward BTK and CRBN but also the absolute concentrations of BTK, CRBN, and PROTAC in cells. To the best of our knowledge,

previous studies on PROTACs only focus on the biochemical measurements for binding affinities of PROTACs to target proteins and E3 ligases without knowing the intracellular concentrations of these proteins.

We performed quantitative western blots to measure the levels of BTK and CRBN in MOLM-14 cells using their corresponding recombinant proteins as the standards. Lysate from a million MOLM-14 cells usually produces 250 µg of protein as determined by BCA assays. Because proteins usually occupy 20–30% of cell volume (assuming 25% w/v, i.e., 250 mg mL$^{-1}$)[34], we can calculate the volume of 1 million MOLM-14 cells to be ~1 µL (i.e., the volume of each MOLM-14 cell is ~1000 µm$^3$). Based on quantitative western blots, we determined the total amount of BTK and CRBN in 1 million MOLM-14 cells to be 60 and 0.72 ng, respectively. Considering the molecular weights of BTK and CRBN are 77 kD and 55 kD, respectively, we can deduce the absolute concentrations for BTK and CRBN in MOLM-14 cells are 780 and 13 nM, respectively (Fig. 3a and Supplementary Fig. 3). Based on our biochemical binding assays, the $K_d$ values between PROTACs and BTK are in the range of 3.0–6.4 nM (Table 1), while the $K_d$ values between the PROTACs and CRBN are in the range of 1.5–4.2 µM (Table 2). Assuming there is no cooperativity effect in the formation of the ternary complex {BTK-PROTAC-CRBN}, we can predict that BTK PROTACs binding to CRBN is the determining factor for ternary complex formation due to the low abundance of CRBN and the weak binding between pomalidomide and CRBN. Therefore, under this situation, further increasing PROTAC binding affinities to BTK, such as comparing RC-1 and RNC-1, would not lead to a

**Table 2 CRBN binding and in-cell target engagement.**

| Compound | $K_d$ (μM)[a] | $IC_{50,TE}$ (μM)[b] | $K'_{P,D}$[c] |
|---|---|---|---|
| RC-1 | 4.2 | 0.25 | 16.8 |
| IRC-1 | 1.5 | 0.86 | 1.7 |
| RNC-1 | 1.8 | 1.69 | 1.1 |
| Pomalidomide | 1.4 | 0.30 | 4.7 |

[a]The dissociation equilibrium constant $K_d$ was measured using a truncated Cereblon protein. Triplicates were performed. For more information please see Supplementary Fig. 11.
[b]CRBN target engagement $IC_{50}$ is the concentration of an unlabeled compound that results in a half-maximal inhibition binding of the CRBN tracer. The target engagement of compounds was assessed following Promega's assay protocol. Triplicates were performed.
[c]$K'_{P,D}$ is a relative intracellular accumulation coefficient for drug D and calculated as $K_d/IC_{50,TE}$. $K'_{P,D}$ is an assay-dependent parameter to quantify the tendency of intracellular accumulation of a drug. Under the same assay conditions, a greater $K'_{P,D}$ value for a drug reflects its higher tendency to accumulate inside cells. Please refer to the Method section for detailed explanation.

meaningful increase of ternary complex formation. This conclusion is also supported by the mathematical model for three-body binding equilibria[35] (Supplementary Fig. 4).

**SPPIER imaging of ternary complex formation in live cells**. To visualize small molecule-induced protein–protein interactions (PPIs), we recently applied fluorophore phase transition-based principle and designed a PPI assay named separation of phases-based protein interaction reporter (SPPIER)[36]. A SPPIER protein design includes three domains, a protein-of-interest, an enhanced GFP (EGFP), and a homo-oligomeric tag (HOTag). Upon small molecule-induced PPI between two proteins-of-interest, multivalent PPIs from HOTags drive EGFP phase separation, forming brightly fluorescent droplets[36]. Here, to detect PROTAC-induced PPI between BTK and CRBN, we engineered the kinase domain of BTK (amino acid residues 382–659, referred to as BTK$^{KD}$) into SPPIER to produce a BTK$^{KD}$-EGFP-HOTag6 construct, which forms tetramers when expressing in cells (Fig. 4a). The previously reported CRBN-EGFP-HOTag3 fusion construct[36], which forms hexamers in cells, was used as the E3 ligase SPPIER (Fig. 4a). If PROTACs can induce {BTK-PROTAC-CRBN} ternary complex formation in cells, they will crosslink the BTK$^{KD}$-EGFP-HOTag6 tetramers and the CRBN-EGFP-HOTag3 hexamers to produce EGFP phase separation, which can be conveniently visualized with a fluorescence microscope. This assay is named as BTK-SPPIER. HEK293T/17 cells were transiently transfected with both constructs. Twenty-four hours after transfection, the cells were incubated with 10 μM of RC-1, IRC-1 or RNC-1. Live cell fluorescence imaging revealed that RC-1, but not IRC-1 nor RNC-1, induced appreciable green fluorescent droplets (Fig. 4b). This imaging result indicated that RC-1 is more efficient to induce {BTK-PROTAC-CRBN} ternary complex formation in living cells than RNC-1 and IRC-1 under the same experimental conditions. It should be noted that the concentration needed for RC-1 to induce appreciable droplet formation in this assay is much higher than its DC$_{50}$ in MOLM-14 cells potentially due to the high overexpression of the target proteins and the sensitivities of the assay.

**Inhibition of cell viabilities by BTK degraders**. We next examined the potencies of inhibiting cell growth for RC-1, IRC-1, RNC-1, RC-1-Me (RC-1 non-degrader control), and their corresponding BTK binder controls, RC-Ctrl, IRC-Ctrl (i.e. ibrutinib), and RNC-Ctrl in MOLM-14 cells. All the chemical structures and IC$_{50}$ values can be found in Fig. 2a and Table 1, respectively. Differing by a methyl group, RC-1 can but RC-1-Me cannot degrade BTK in cells (Fig. 2d). Interestingly, both compounds have similar IC$_{50}$ values (0.31 vs 0.21 μM), suggesting that BTK inhibition but not degradation accounts for the toxicity in

MOLM-14 cells. The IC$_{50}$ values for RC-Ctrl, IRC-Ctrl, and RNC-Ctrl are also similar in the range of 0.3–0.5 μM, suggesting that these three warheads inhibit BTK to a similar extent in cells. Surprisingly, both the IC$_{50}$ values for IRC-1 and RNC-1 are in the μM range (2.7 and 4.1 μM). A biochemical BTK kinase inhibition assay showed that IRC-1 and RNC-1 have slightly diminished inhibitory activities (<3-fold) compared with their corresponding warhead controls (Table 1). However, the difference between enzymatic activities is insufficient to explain the >10-fold difference in IC$_{50}$ values in cells, suggesting that IRC-1 and RNC-1 may have poorer intracellular accumulation than their corresponding warhead controls. In contrast, RC-1 and RC-Ctrl have similar IC$_{50}$ values in both biochemical BTK inhibition and cellular growth inhibition assays (Table 1), suggesting similar compound exposure in cells. The growth inhibition assay for the BTK degraders prompted us to investigate alternative mechanisms to explain the high potency of RC-1.

**Comparison of intracellular accumulation of BTK degraders**. Since the binding affinities between BTK and its PROTACs cannot explain the difference in BTK degradation between RC-1 and RNC-1 and potencies for cell growth inhibition among RC-1, RNC-1, and IRC-1, we asked whether the intracellular concentration of RC-1 may be higher than those of RNC-1 and IRC-1, leading to more potent pharmacological effects. To test this possibility, we turned to the Nano-luciferase (nLuc) based bioluminescence resonance energy transfer (NanoBRET) assay[37]. HEK-293 cells were transiently transfected with plasmids expressing a fusion protein of Cereblon and nano-luciferase for 24 h and then the cells were treated with a Cereblon tracer, which binds to Cereblon to induce NanoBRET signals. Adding PROTACs to cells would compete the CRBN tracer binding to CRBN, thus reducing the NanoBRET signals. The target engagement IC$_{50}$ is defined as the concentration of an unlabeled compound that results in a half-maximal inhibition binding between the fluorescent tracer and the nanoLuc fusion protein[38]. The CRBN target engagement IC$_{50}$ value of RC-1 is 3 and 7-fold as low as those of IRC-1 and RNC-1, respectively (Fig. 3b). It should be noted that target engagement IC$_{50}$ values are dependent on assay conditions, including the tracer concentration and the expression level of the nanoLuc fusion protein. Therefore, the target engagement IC$_{50}$ values for different compounds can only be meaningfully compared under the same assay conditions.

To quantitatively compare the intracellular concentrations of these PROTACs, we defined a parameter $K_{P,D}$ as the intracellular accumulation coefficient for drug D, in which P and D denote partition and drug, respectively, following the previous work (see Methods section for details)[39]. $K_{P,D}$ is defined as the ratio between the total intracellular and extracellular concentrations of drug D and calculated as $K_{P,D} = \frac{C_{D,in}}{C_{D,ex}}$, where $C_{D,in}$ and $C_{D,ex}$ are the total intracellular and extracellular concentrations of drug D, respectively. When 50% of target engagement is reached with the addition of drug D to the medium (i.e. $C_{D,ex} = IC_{50,TE}$ assuming intracellular drug accumulation does not significantly change the extracellular drug concentration; $C_{D,in} = [D]$ assuming the total intracellular concentration of drug D is much greater than that of the target protein P), we can deduce that $K_{P,D} = \frac{C_{D,in}}{C_{D,ex}} = \frac{[D]}{IC_{50,TE}} = \frac{K_{d,D}}{IC_{50,TE}} \frac{[PD]}{[P]}$, where $K_{d,D}$ is the dissociation equilibrium constant for the binding between the target protein or its nanoLuc fusion and the drug, [P] is the equilibrium concentration of the target protein nanoLuc fusion, [D] is the equilibrium concentration of the free drug D, [PD] is the equilibrium concentration of the drug bound form of the target protein nanoLuc fusion, and IC$_{50,TE}$ is the target engagement IC$_{50}$ in the

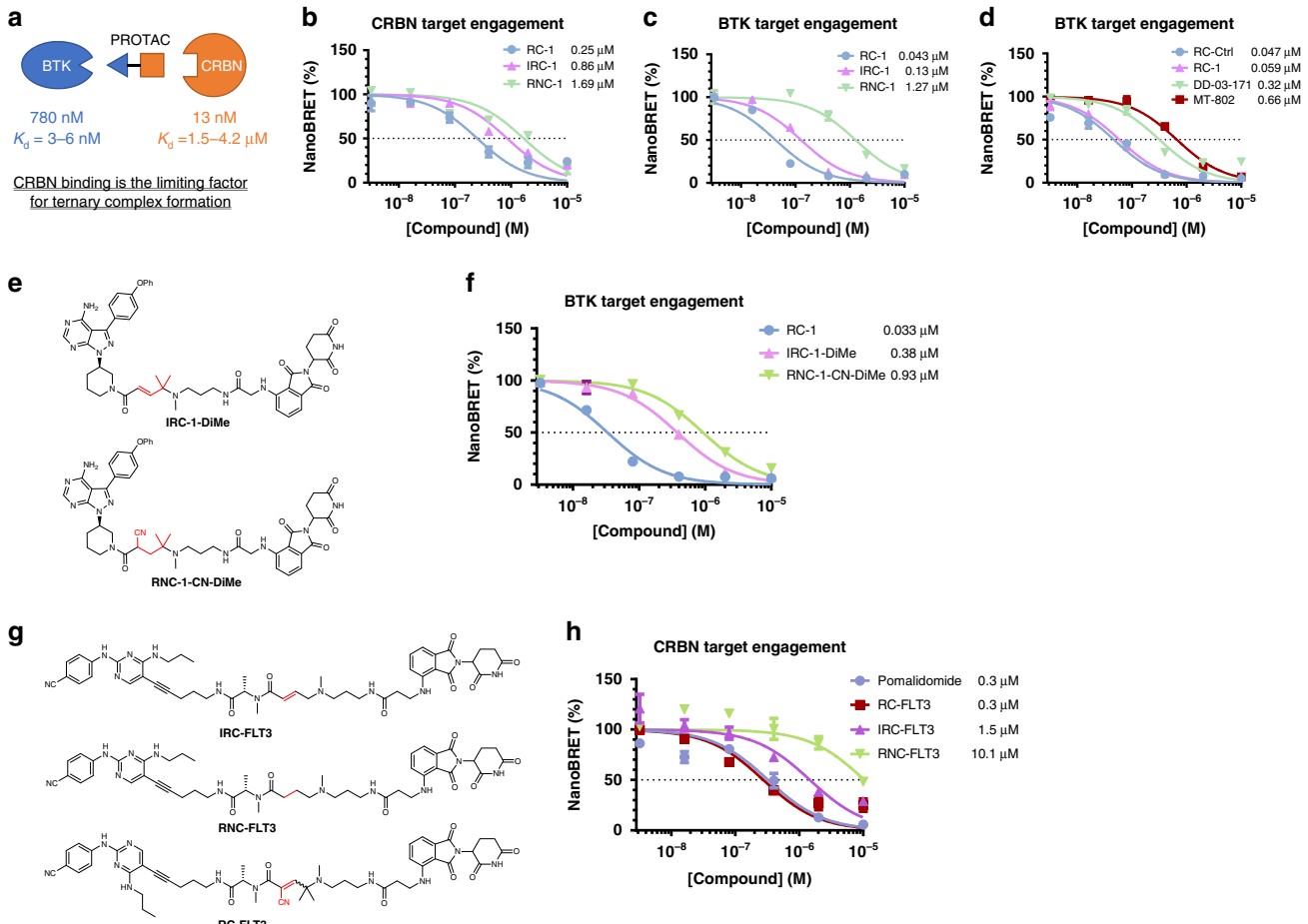

**Fig. 3 Target engagement for PROTACs in cells. a** Based on quantitative western blots (Supplementary Fig. 3) and assuming the MOLM-14 cell volume is ~1000 μm³, BTK and CRBN concentrations are 780 and 13 nM, respectively. The weak binding between pomalidomide and CRBN suggests that CRBN binding is the limiting factor for ternary complex formation. **b** CRBN in-cell target engagement assay. HEK-293 cells were transiently transfected with plasmids expressing a fusion protein of CRBN and nano-luciferase (nLuc) for 24 h and then the cells were treated with a CRBN tracer (0.5 μM), which binds to CRBN to induce bioluminescence resonance energy transfer (BRET). Adding PROTACs to cells would compete the CRBN tracer binding to CRBN, thus reducing the NanoBRET signals. It should be noted that the NanoBRET assay is ratiometric and independent of the expression level of the nLuc fusion protein. The target engagement $IC_{50}$ values for RC-1 (blue circle), IRC-1 (violet pyramid), and RNC-1 (green inverted pyramid) are 0.25, 0.86, and 1.69 μM, respectively. **c** BTK in-cell target engagement assay. This assay is the same as in **b**, except BTK-nLuc fusion plasmid and BTK tracer (1.0 μM) were used. The target engagement $IC_{50}$ values for RC-1 (blue circle), IRC-1 (violet pyramid), and RNC-1 (green inverted pyramid) are 0.043, 0.13, and 1.27 μM, respectively. **d** The same BTK in-cell target engagement assay as in **c** was applied to RC-Ctrl (blue circle), RC-1 (violet pyramid), DD-03-171 (green inverted pyramid), and MT-802 (brown square). The target engagement $IC_{50}$ values for RC-1 and RC-Ctrl are the same within experimental errors, demonstrating that the intracellular accumulation of RC-1 is similar to its parent warhead molecule. In contrast, the target engagement $IC_{50}$ values for DD-03-171 and MT-802 are 5 and 11 folds of that of RC-1, respectively. Triplicates were performed with SEM as the error bars. **e** Chemical structures of IRC-1-DiMe and RNC-1-CN-DiMe. **f** BTK target engagement for RC-1 (blue circle), IRC-1-DiMe (violet pyramid), and RNC-1-CN-DiMe (green inverted pyramid). The same BTK in-cell target engagement assay as in **c** was applied to RC-1, IRC-1-DiMe, and RNC-1-CN-DiMe. The $IC_{50}$ values for RC-1, IRC-1-DiMe, and RNC-1-CN-DiMe are 0.033, 0.38, and 0.93 μM, respectively. **g** Chemical structures of FLT3 degraders. **h** CRBN in-cell target engagement assay as in **b**. The target engagement $IC_{50}$ values for pomalidomide (blue circle), RC-FLT3 (brown square), IRC-FLT3 (violet pyramid), and RNC-FLT3 (green inverted pyramid) are 0.3, 0.3, 1.5, and 10.1 μM, respectively. Data are presented as mean values ± SEM ($n = 3$ biologically independent samples). Source data are provided as a Source Data file.

NanoBRET assay. Under the same assay conditions, [PD] and [P] are constants and [PD]/[P] can be defined as a constant A. So, we can deduce that $K_{P,D} = A \frac{K_{d,D}}{IC_{50,TE}}$. We further define the relative intracellular accumulation coefficient for drug D ($K'_{P,D}$) as $K'_{P,D} = \frac{K_{P,D}}{A} = \frac{K_{d,D}}{IC_{50,TE}}$. Under the same assay conditions, a greater $K'_{P,D}$ value for a drug reflects its higher tendency to accumulate inside cells. It should be noted that $K_{P,D}$ is independent of assays and assay conditions but requires quantification of [P] and [PD]. In contrast, $K'_{P,D}$ is an assay condition dependent parameter but

provides a convenient approach to quantitatively compare the tendency of intracellular accumulation of drugs.

Based on the $K'_{P,D}$ values calculated by dividing the $K_d$ to CRBN with the in-cell target engagement $IC_{50}$ (Table 2), we can deduce that the intracellular accumulation of RC-1 is 10 and 16-fold as the levels of IRC-1 and RNC-1, respectively. Additionally, the calculated cLogP and polar surface area (PSA) values for these three compounds are similar (Supplementary Table 1), arguing against the possibility that the physicochemical properties of RC-1 are the cause of its high intracellular concentration. Therefore,

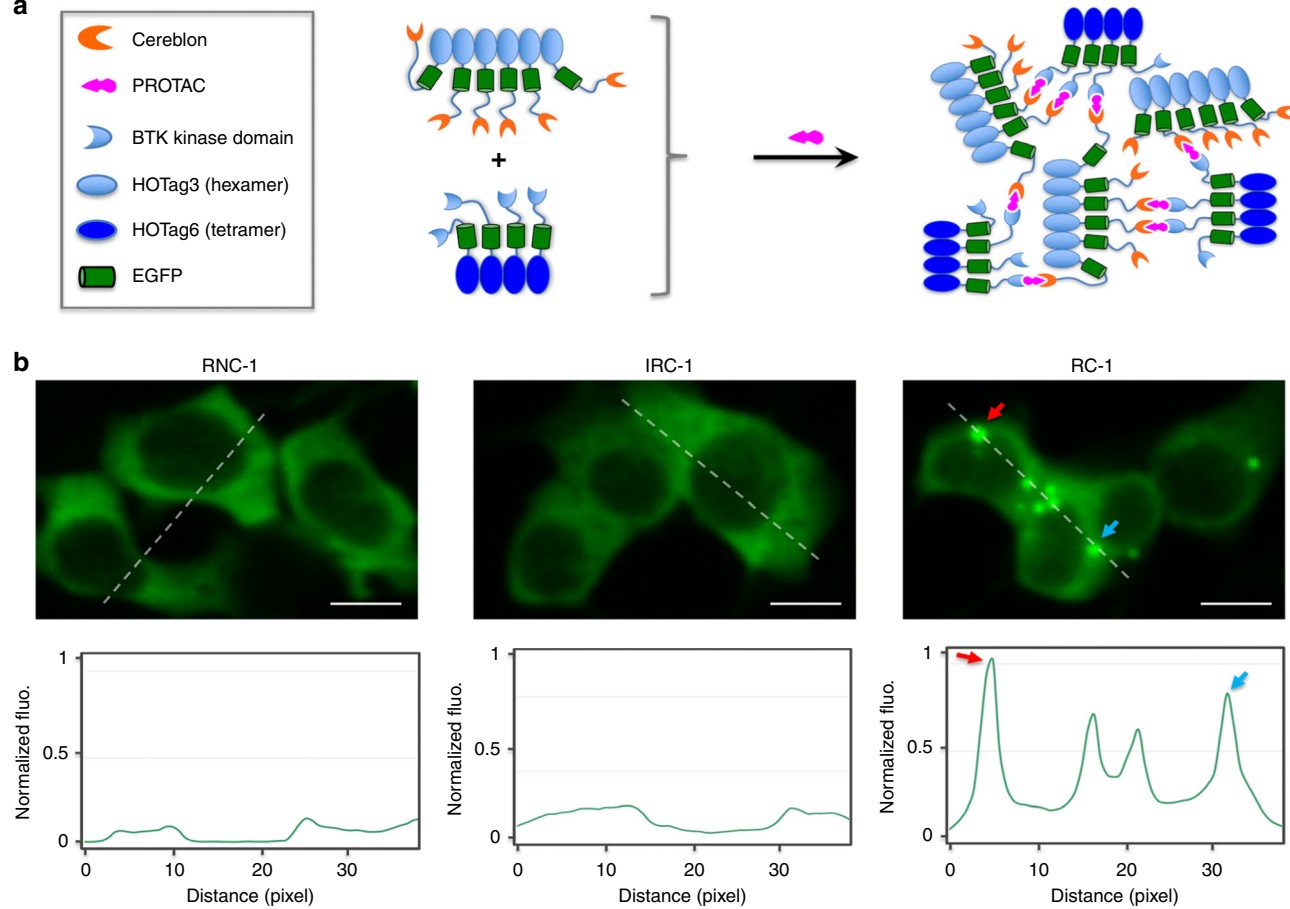

**Fig. 4 Fluorophore phase separation-based assay for imaging ternary complex formation in living cells. a** Schematic diagram showing the design of the cellular assay. To detect PROTAC-induced PPI between BTK and CRBN, we engineered the kinase domain of BTK (amino acid residues 382–659, referred to as BTK$^{KD}$) into SPPIER to produce a BTK$^{KD}$-EGFP-HOTag6 construct, which forms tetramers when expressing in cells. The previously reported CRBN-EGFP-HOTag3 fusion construct, which forms hexamers in cells, was used as the E3 ligase SPPIER. If PROTACs can induce {BTK-PROTAC-CRBN} ternary complex formation in cells, they will crosslink the BTK$^{KD}$-EGFP-HOTag6 tetramers and the CRBN-EGFP-HOTag3 hexamers to produce EGFP phase separation, which can be conveniently visualized with a fluorescence microscope. **b** Fluorescence images showing detection of BTK PROTACs-induced interaction between the E3 ligase cereblon and the target protein BTK. A fluorescence histogram of the line across the cells is shown below. HEK293 cells transiently expressed BTK$^{KD}$-EGFP-HOTag6 and CRBN-EGFP-HOTag3. RC-1, IRC-1, and RNC-1 (10 μM) were added to the cells. Scale bar: 10 μm. Source data are provided as a Source Data file.

we conclude that the efficient BTK degradation and potent cell growth inhibition induced by RC-1 is achieved mostly through its high intracellular accumulation, due to the reversible covalent structural moiety in RC-1.

**RC-1 is a unique BTK degrader with high target occupancy.** Although the PROTACs are characterized as BTK degraders, they have warheads that can bind and inhibit BTK, essentially as dual-functional BTK inhibitors and degraders. We performed biochemical BTK kinase inhibition assays to measure the IC$_{50}$ values for RC-1, RNC-1, and IRC-1 and their corresponding warhead controls (Table 1 and Supplementary Fig. 5). IRC-Ctrl (i.e. Ibrutinib) forms a covalent bond with BTK and is expected to be the most potent BTK inhibitor (IC$_{50}$ = 0.3 nM). In comparison, RC-Ctrl and RNC-Ctrl have reduced BTK inhibition activities by 7 and 45 folds, respectively. The BTK PROTACs, RC-1, RNC-1, and IRC-1, have similar BTK inhibitory activities to their corresponding warheads (Table 1).

We then performed a similar NanoBRET-based live-cell target engagement assay for BTK. Consistent with the CRBN target engagement assay, we found that the BTK target engagement IC$_{50}$ value of RC-1 is 3 and 30 folds of the values for IRC-1 and

RNC-1, respectively (Fig. 3c). Based on the $K'_{P,D}$ values calculated by dividing the $K_d$ to BTK with the in-cell target engagement IC$_{50}$ (Table 1), we can deduce that the intracellular accumulation of RC-1 is 6 and 40 folds as high as those of IRC-1 and RNC-1, respectively, similar to the trend observed in the CRBN target engagement assay (Fig. 3b).

To further rule out the possibility that the enhanced intracellular accumulation and target engagement of RC-1 is due to its physical properties, we synthesized two additional control compounds RNC-1-CN-DiMe and IRC-1-DiMe (Fig. 3e). RNC-1-CN-DiMe can be viewed as direct reduction of the C=C double bond in the Michael acceptor of RC-1 to a single bond. IRC-1-DiMe can be viewed as RC-1 only lacking the cyano group but maintaining the dimethyl moiety. The BTK target engagement IC$_{50}$ value of RC-1 is at least an order of magnitude smaller than the values for IRC-1-DiMe and RNC-1-CN-DiMe, respectively (Fig. 3f). Calculating the $K'_{P,D}$ values, we can deduce that the intracellular accumulation of RC-1 is 5 folds of those of IRC-1-DiMe and RNC-1-CN-DiMe (Table 1), arguing against that the enhanced intracellular accumulation of RC-1 is attributed to the physical property changes caused by the additional cyano or dimethyl groups in RC-1 compared with IRC-1 and RNC-1.

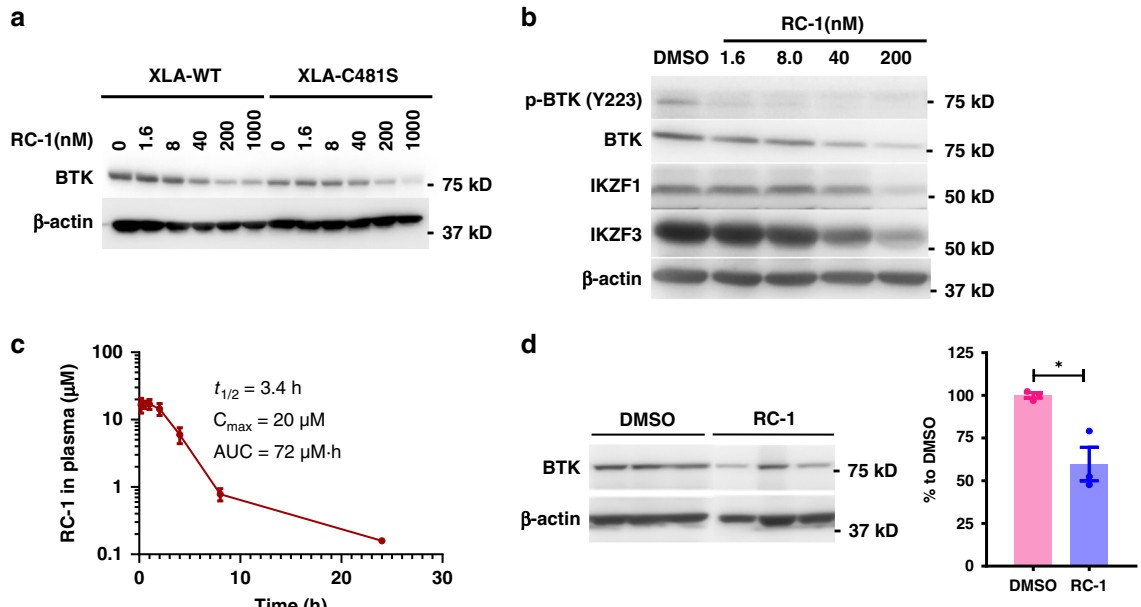

**Fig. 5 In vitro and in vivo efficacy of RC-1. a** XLA cells overexpressing wild-type or C481S mutant BTK were treated with RC-1 for 24 h. Then western blotting was performed to evaluate the degradation of BTK. Duplicates were performed. **b** RC-1 induced degradation of BTK, pBTK, IKZF1, and IKZF3 in mino cells. Duplicates were performed. **c** Pharmacokinetics of RC-1 in mice. RC-1 (20 mg kg$^{-1}$) was injected intraperitoneally in ICR mice ($n = 3$) and the plasma was analyzed using LC-MS/MS. Data are presented as mean values ± SD ($n = 3$ mice per group). **d** Representative western blot (left) and quantification (right) of splenic BTK level in mice ($n = 3$) treated with 7 once daily i.p. injection of 20 mg kg$^{-1}$ RC-1 (blue bar, spleens were harvested 24 h after the 7th injection) and DMSO (violet bar). Data are presented as mean values ± SEM ($n = 3$ biologically independent animals). Asterisks indicate that the differences between samples are statistically significant, using two-tailed, unpaired $t$-test (*$P < 0.05$). Source data are provided as a Source Data file.

Due to poor intracellular accumulation, most PROTACs have low target occupancy (Fig. 3d) and rely on the sub-stoichiometric protein degradation to achieve maximal efficacy. Using the NanoBRET-based BTK in-cell target engagement assay, we found that RC-1 can achieve 50 and 90% of target engagements at 40 and 200 nM, respectively. Therefore, RC-1 can function as both a BTK inhibitor and degrader.

**RC-1 degrades BTK regardless of its mutation status.** Mutations in BTK, C481S in particular, confer ibrutinib resistance in clinic[26]. We set out to test whether BTK degradation induced by our PROTACs is affected by BTK mutation status. XLA cells overexpressing wild-type BTK or C481S mutant BTK were treated with RC-1, RNC-1, and IRC-1 for 24 h, followed by western blot to compare the BTK levels (Fig. 5a and Supplementary Fig. 6). We observed dose-dependent BTK degradation induced by RC-1 regardless of its mutation status with comparable potency. This observation is consistent with our previous conclusion that altering PROTAC binding affinities to BTK within a range does not significantly change the ternary complex formation efficiency (Supplementary Fig. 4). It should be noted that the potency of RC-1 is weaker in XLA cells than in MOLM-14 cells possibly because BTK is overexpressed in XLA cells. It is also interesting to note that IRC-1 induces much more effective degradation of the BTK C481S mutant than its wild-type in XLA cells (Supplementary Fig. 6), suggesting that the irreversible covalent bond formation between IRC-1 and BTK causes the inefficient protein degradation, consistent with the previous study by the GSK group[15].

**RC-1 degrades BTK with higher specificity than IRC-1 and RNC-1.** To explore the effects of our degraders on the whole proteome, we treated MOLM-14 cells with RC-1, RNC-1, IRC-1,

RC-1-Me (non-degrader control) or DMSO and employed a quantitative multiplexed proteomic approach to measure the whole cellular protein levels (Fig. 6). The result showed that both in IRC-1 and RNC-1-treated cells, 7 kinases are degraded including BTK. However, for RC-1-treated cells, only two kinases (BTK and CSK) can be degraded, suggesting that RC-1 has more selectivity than IRC-1 and RNC-1 for kinase degradation. However, no degradation is observed in RC-1-Me-treated cells, indicating that the degradation observed for RC-1 is CRBN dependent. In addition, immunomodulatory imide drugs (IMiD)-dependent substrates including IKZF1, ZFP91, and ZNF692 are also specifically degraded by RC-1, RNC-1, and IRC-1.

**MD simulations predict a stable ternary complex for RC-1.** To better understand the ternary complex formation, we applied molecular dynamics simulations to evaluate the global rearrangement of the ternary structures for BTK, CRBN, and RC-1 or IRC-1 or RNC-1, and compared the binding stabilities of these three ligands to BTK and CRBN. The residue-based RMSF (root mean square fluctuations) was calculated based on the entire simulation trajectory (>200 ns) to assess the average fluctuation of each residue of the three simulated complexes. It is clear that RNC-1 mediated complex {BTK-RNC-1-CRBN} has the largest fluctuation compared with the other two complexes {BTK-RC-1-CRBN} and {BTK-IRC-1-CRBN} (Fig. 7a), suggesting that covalent binding could help to overcome the unfavorable entropy penalties during the ternary complex formation. Additionally, the RMSD (Root-mean-square deviation) values were calculated based on the Cα atoms to reflect the conformation changes from the input structures based on x-ray crystallography. Consistent with the RMSF analyses, the RMSD changes for the {BTK-RC-1-CRBN} and {BTK-IRC-1-CRBN} complexes are much smaller than the one for the {BTK-RNC-1-CRBN} complex, suggesting

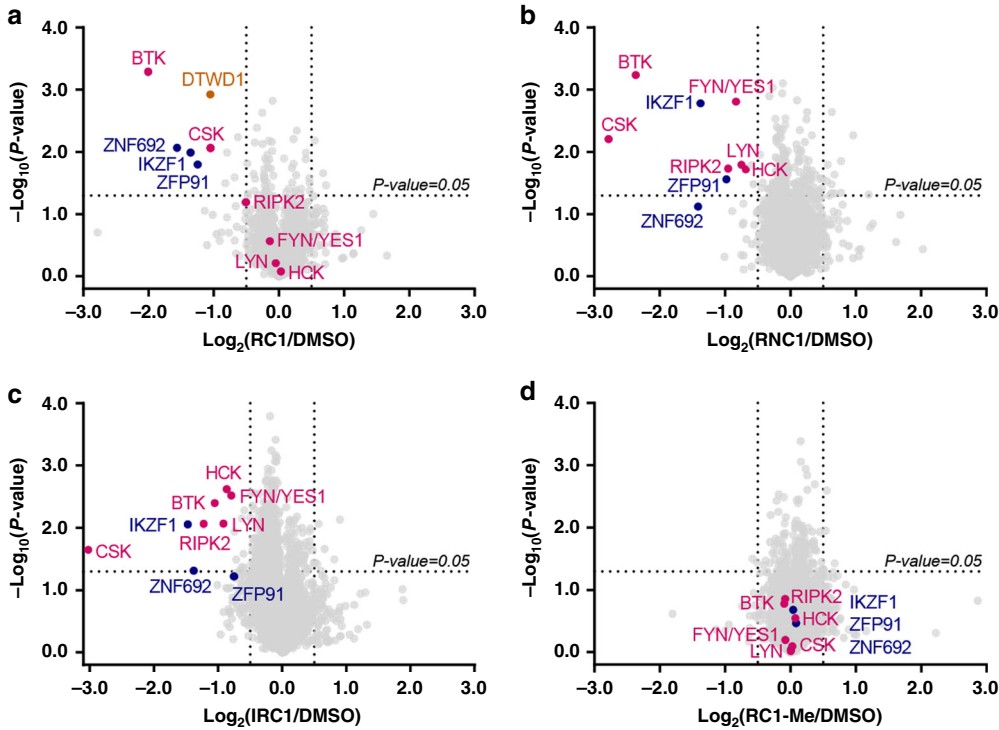

**Fig. 6 Proteomic analysis showing RC-1 selectively degrades BTK.** MOLM-14 cells were treated with either compounds (200 nM **a** RC-1, **b** RNC-1, **c** IRC-1, or **d** RC-1-Me (RC-1 non-degrader control)) or DMSO for 24 h. Lysates were treated with a TMT-10plex kit and subjected to mass spec-based proteomics analysis. Datasets represent an average of duplicates. Volcano plot shows protein abundance ($\log_2$) as a function of significance level ($\log_{10}$). Nonaxial vertical lines denote abundance changes from 0.7 to 1.4 (i.e. $2^{\pm 0.5}$), whereas nonaxial horizontal line marks $P = 0.05$ significance threshold. Downregulated proteins of significance are found in the upper left quadrant of the plots. Violet red dots represent kinases. Dark blue dots represent zinc-finger proteins. Both RNC-1 and IRC-1 showed significant degradation of 7 kinases, while RC-1 only degrades BTK and CSK. A total of 7280 proteins were identified, and only the ones with at least one uniquely identified peptide are displayed. The raw data are provided as Supplementary Data 2.

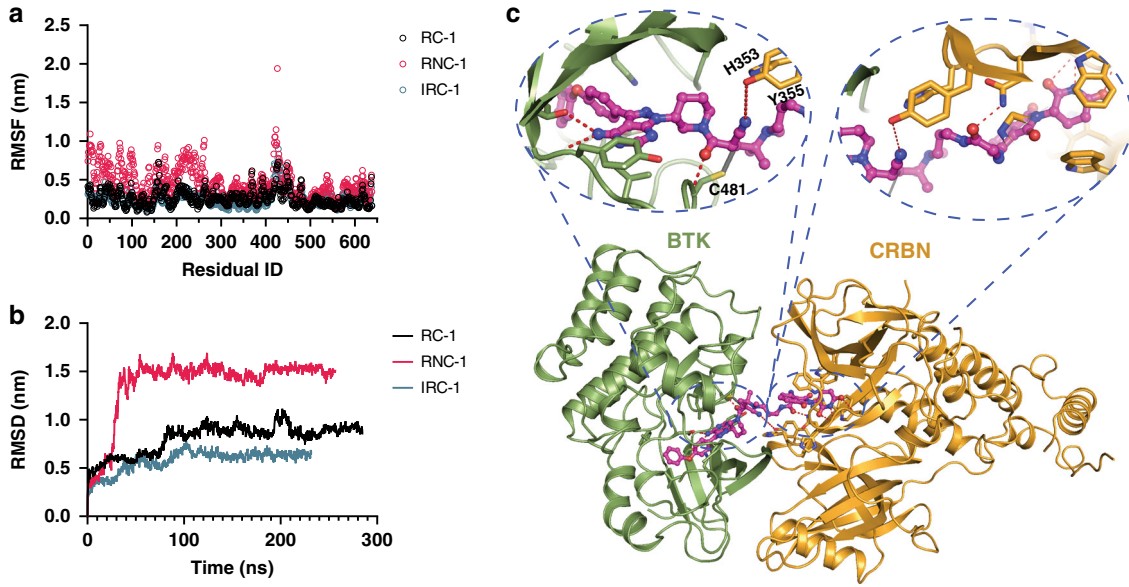

**Fig. 7 Molecular dynamics simulations for ternary complexes of BTK-PROTAC-CRBN.** RNC-1 mediated complex shows a larger structural fluctuation compared with the other two ligands, reflected by **a** Root Mean Square Fluctuation (RMSF) per residue and **b** Root Mean Square Deviation (RMSD) calculated from the simulations of the {BTK-PROTAC-CRBN} ternary complexes. **c** Predicted most stable conformation of {BTK-RC-1-CRBN} ternary complex. Red dashes indicate the hydrogen bonds formed among the ligand, proteins, or waters. Light green cartoons are BTK and cyan cartoons are E3 ligase. Magenta stick ball models represent RC-1.

that covalent binding can help to stabilize the ternary complexes (Fig. 7b).

Low energy conformations of the protein complex can shed light on the detailed molecular interactions which stabilize the binding interface. The conformation with the lowest energy along the trajectories was extracted and shown in Fig. 7c. The binding mode and critical molecular interactions, i.e., hydrogen bonds and $\pi$–$\pi$ interactions formed between the ligands and BTK and CRBN, are conserved compared to the crystal structure of BTK-ibrutinib and CRBN-lenalidomide. From the predicted binding mode, several hydrogen bonds and the covalent bond between RC-1 and the residue C481 of BTK anchors the ligand to the binding site of BTK. Additionally, a hydrogen bond between Y355 of CRBN and anion-$\pi$ interaction formed between H353, hence, further greatly stabilize the binding of RC-1 with proteins, and help to hold the orientation of RC-1 to adjust and stabilize the ternary complex (Fig. 7c).

**Comparison between RC-1 and other reported BTK degraders.** The goal of blocking BTK signaling with either BTK inhibitors or degraders is to inhibit the growth of cancer cells. To this end, we synthesized previously reported BTK degraders DD-03-171[32] and MT-802[31] and compared them head-to-head with RC-1, RNC-1, and IRC-1 in their abilities to inhibit cancer cell growth (Fig. 2f and Supplementary Fig. 7). Due to the structural similarities of RNC-1 and the BTK degraders reported by the GSK group (PROTAC 5)[15] and the Rao group[28], we used RNC-1 as a surrogate for comparison. In MOLM-14 cells, RC-1 has the most potent inhibitory effect among all the PROTACs compared (Fig. 2f). It is interesting to note that RC-1 and RC-1-Me, which does not induce BTK degradation, have similar $IC_{50}$ values in inhibiting MOLM-14 cell growth, indicating that the growth inhibitory effect induced by PROTACs in MOLM-14 cells is due to BTK inhibition instead of its degradation. The high potency of RC-1 is possibly due to the combinatorial effects of its high intracellular concentration and tight binding to BTK.

We also compared the potency of BTK degraders in Mino cells, a BTK dependent mantle cell lymphoma (MCL) cell line. In Mino cells, RC-1 and DD-03-171 have comparable potency for inhibiting cell growth and outperform all the other BTK PROTACs tested (Fig. 2f). Additionally, RC-1 can degrade not only BTK and phosphorylated BTK but also IKZF1 and IKZF3 (Fig. 5b), similar to DD-03-171[32]. It is interesting to note that RC-1 is more potent than ibrutinib in Mino cells, but have similar potencies in Jeko-1, Rec-1, and Maver-1 cells (Supplementary Table 2). Further investigation is required to identify biomarkers for the subtype of MCL sensitive to BTK degraders.

Additionally, we compared BTK in-cell target engagement among RC-1, RC-Ctrl (RC-1 warhead) and these reported BTK degraders (Fig. 3d). It is remarkable to see that the target engagement $IC_{50}$ values for RC-1 and RC-Ctrl (59 vs 47 nM) are the same within experimental errors, demonstrating that the intracellular accumulation of RC-1 is essentially the same as its parent warhead molecule although their molecular weights differ by 1.7-fold. In contrast, the target engagement $IC_{50}$ values for DD-03-171 and MT-802 are 5 and 11 folds as high as that of RC-1, respectively.

**RC-1 has an ideal plasma half-life and degrades BTK in vivo.** We measured the plasma half-life of RC-1 (20 mg kg$^{-1}$, i.p. injection) in ICR mice (female, 5–6 weeks, $n = 3$) using LC-MS/MS. The PK data were fitted into a non-compartmental model using PK solver[40]. RC-1 has a plasma half-life ($t_{1/2}$) of 3.4 h, $C_{max}$ of 20 μM, and AUC of 72 μM·h (Fig. 5c). To further test the pharmacodynamics of RC-1 in vivo, we treated ICR mice ($n = 3$)

with 20 mg kg$^{-1}$ of RC-1 (i.p. injection). After 7 times of daily injections, the mice were killed, and their spleens, the locations of the majority of B cells, were collected. Western blotting showed that RC-1-treated mice had ~50% BTK level reduction in the spleens compared with the vehicle-treated mice (Fig. 5d and Supplementary Fig. 8). Therefore, this preliminary study showed that RC-1 has desirable PK/PD properties in vivo, laying the foundation for future efficacy tests in mouse models to ultimately move this degrader into the clinic.

**Other reversible covalent PROTACs with improved TE.** Based on the success of RC-1, we were curious whether cyano-acrylamide based reversible covalent PROTACs can be generally applied to improve target engagement. An activating mutation of Fms-like tyrosine kinase 3 (FLT3) is the most frequent genetic alteration associated with poor prognosis in acute myeloid leukemia (AML). Similar to BTK, FLT3 also has a cysteine residue in the ATP binding pocket. Yamaura et al.[41] reported a highly potent irreversible covalent FLT3 inhibitor, termed as FF-10101. Building on this work, we designed a series of reversible covalent, irreversible covalent, and reversible non-covalent FLT3 PROTACs. Using the NanoBRET CRBN in-cell target engagement assay, we found that RC-FLT3 has a similar target engagement $IC_{50}$ value compared with that for pomalidomide (Fig. 3h). Similar to BTK PROTACs, the target engagement $IC_{50}$ values for IRC-FLT3 and RNC-FLT3 PROTACs are 5 and 34 folds as high as that for RC-FLT3 (Fig. 3g, h). This experiment confirmed that introducing a cyano-acrylamide moiety can be a general strategy to enhance intracellular accumulation and target engagement for PROTACs.

**Discussion**
Despite the exciting promise of PROTACs, the development of this type of therapeutic agent faces some major challenges. Due to the high molecular weights of PROTACs, they are not druglike based on the commonly applied empirical rules for drug discovery and tend to have poor membrane permeabilities, resulting in low drug concentrations inside cells. Therefore, PROTACs usually have low target occupancy and heavily depend on sub-stoichiometric protein degradation to achieve therapeutic efficacy. Although there are successful PROTAC examples with potent in vitro and in vivo efficacies, they are mostly optimized through brute force medicinal chemistry efforts. Few efforts have been focused on developing general strategies to improve the intracellular accumulation of PROTACs.

We serendipitously discovered that cyano-acrylamide groups enhance intracellular accumulation of PROTACs. We prefer the term enhanced intracellular accumulation instead of enhanced permeability to describe the unique feature of RC-1. Permeability characterizes the potential for a compound to transverse the plasma membrane of cells, which can be energy independent or dependent. In contrast, intracellular accumulation is a general term to describe the extent of an exogenous compound remains inside cells, which is an agglomerate effect determined by permeability, specific, and non-specific interactions with intracellular proteins and other biomolecules, and elimination through metabolism and drug efflux pumps.

Based on Taunton's and our own previous work[22,42,43], cyano-acrylamides can reversibly react with thiols with mM dissociation equilibrium constants ($K_d$) and rapid kinetics. It is possible that the enhanced intracellular accumulation of RC-1 is due to its fast and reversible reactions with intracellular glutathione (~1–10 mM), serving as a sink to trap RC-1 in cells. In contrast, RNC-1 cannot react with GSH. Technically, IRC-1 can irreversibly react with GSH but with a much slower reaction rate. This hypothesis

could be tested by measuring the target engagement $IC_{50}$ values of RC-1 at different intracellular GSH concentrations in HEK293 cells. Additionally, it is also possible that many available free cysteine residuals on the cellular surface can reversibly react with cyano-acrylamide containing molecules to mediate the enhanced cellular uptake. The Matile group reported that strained cyclic disulfides can enhance cellular uptake of non-permeable molecules through a similar mechanism[44,45]. Although understanding the molecular mechanism of enhanced intracellular accumulation of RC-1 is out of the scope of this work, it would be interesting to explore other reversible covalent chemistry to enhance the cellular uptake of poorly permeable molecules.

We attempted to measure the permeabilities of RC-1, IRC-1, and RNC-1 using the lipid-PAMPA method (Supplementary Table 3 and Supplementary Data 3). Unfortunately, none of the compounds were detected on the receptor side, indicating extremely poor physical permeabilities. Additionally, the total recovery rates for all the three compounds were only in the range of 20–30%, suggesting that these compounds may stick to the PAMPA lipids and/or the plasticware used in this assay. Therefore, we concluded that the lipid-PAMPA method is not an ideal assay to evaluate the permeabilities of our PROTACs.

We developed the concept of relative intracellular accumulation coefficient $K'_{P,D}$ to quantitatively compare the intracellular concentrations of drugs under the same assay conditions, taking advantage of the live-cell-based NanoBRET assays. Initially, we attempted to use LC-MS to quantify the intracellular concentrations of the PROTACs by extracting them from the cell lysates (Supplementary Table 4 and Supplementary Data 4). Although we found that RC-1 has slightly higher concentrations (~2-fold) than RNC-1 and IRC-1 inside cells, it is difficult to differentiate whether the drug molecules measured are indeed intracellular or non-specifically attach to the cell surface using the lysate method. Considering the greasy nature of the PROTACs revealed in the lipid-PAMPA permeability experiment, we are concerned about the accuracy of the lysate-based drug quantification in cells. In contrast, the live-cell-based NanoBRET assays are less prone to the artefacts associated with lysate-based methods and produce more reliable comparison of target engagement and intracellular concentration of PROTACs. Combining the target engagement $IC_{50}$ values in the NanoBRET assays and the dissociation equilibrium constants ($K_d$) between the drug and the target protein, we can calculate the relative intracellular accumulation coefficient $K'_{P,D}$ as a convenient and reliable method to compare the tendency of drugs accumulating in cells.

Despite RC-1 is highly potent to degrade BTK in human cell culture and has a favorable PK property in mice, the maximum BTK degradation achieved in splenic B cells in mice is only ~50% even with seven daily injections. To reconcile the discrepancy, we treated a mouse B-cell lymphoma cell line derived from Eu-Myc mice with RC-1 in vitro and found that the maximum BTK degradation is only 30–40% even dosed up to 25 μM of RC-1 (Supplementary Fig. 9), indicating that RC-1 is indeed much less potent for BTK degradation in mouse cells than human cells. Although thalidomide and its analogs are clinically effective treatments for multiple myeloma, they are inactive in murine models due to the sequence differences between mouse and human Cereblon[46,47]. Mice with a single I391V mutation in Crbn restored thalidomide-induced degradation of drug targets previously found in human cells[46,47]. We rationalize that the BTK degradation potency difference for RC-1 in human and mouse cells may be due to the sequence variations for BTK and/or CRBN in different species, leading to inefficient formation of the ternary complex among RC-1, mouse BTK, and mouse CRBN due to altered protein–protein interactions. This raised a question whether mice with humanized CRBN, either I391V and/or other

mutations, should be developed to properly access the efficacy and toxicity of CRBN recruiting PROTACs in murine models.

The efficacy of irreversible covalent PROTACs has been controversial, with both successful[7,16–19] and unsuccessful examples[15]. An apparent drawback for irreversible covalent PROTACs is its non-catalytic degradation of target proteins. If we examine the target engagement data for IRC-1 (Fig. 3b, c), it can achieve ~100% BTK and ~40% CRBN engagement at 400 nM of IRC-1. Assuming the target engagement $IC_{50}$ values are similar in the HEK293 overexpression system and MOLM-14 cells, we would expect to observe significant BTK degradation with 400 nM of IRC-1 treatment, even with the single turnover degradation. However, very little BTK degradation was observed in MOLM-14 cells with IRC-1 treatment, suggesting either BTK was not efficiently ubiquitinated due to the poor orientation of the ternary complex, or ubiquitinated BTK was not effectively degraded by proteasomes. The detailed mechanism to explain why IRC-1 cannot efficiently degrade BTK requires further investigations. Nonetheless, RC-1, the reversible covalent counterpart of IRC-1, addresses the non-catalytic degradation issue of IRC-1 and brings unexpected intracellular accumulation advantages.

Disrupting the kinase function of BTK using kinase inhibitors is the only currently available therapeutic intervention on this well-validated target. However, it was shown that BTK can enhance antigen receptor-induced calcium influx in a kinase-independent manner[48], while a kinase-inactive BTK mutant can partially rescue B cell development of BTK null B cells in mice[49]. Additionally, conditional mouse knockouts of BTK after the establishment of B-cell populations have shown that B-cell populations were maintained in conditional knockout mice[50], suggesting that BTK may not be essential for the survival of mature, normal human B-cells but is essential for the survival of MCL and CLL cells. In this work, we developed a small molecule therapeutic agent that can efficiently inhibit and degrade BTK irrespective of BTK mutation status to abolish both BTK kinase and non-kinase activities completely. It should be noted that although a few BTK degraders have been reported[15,28,30–32], they have low target engagement due to their poor intracellular accumulation (Fig. 3d). It is usually very difficult for degraders to achieve 100% degradation of the target protein, and the remaining small portion of undegraded BTK can be sufficient to sustain the signaling cascades due to its enzymatic nature. Clinically, it is important for ibrutinib to achieve 100% BTK engagement to completely shut down the BTK downstream signaling. In terms of BTK degradation, RC-1 is more potent than MT-802 at a low concentration (8 nM) and has a comparable potency at a high concentration (200 nM; Supplementary Fig. 10). However, our BTK degrader RC-1 is unique compared with other BTK degraders because it not only degrades BTK efficiently but also shows high target engagement to inhibit BTK in case BTK is not completely degraded. This dual mechanism of action (MOA) for BTK inhibition and degradation deserves further exploration in the clinic.

While this manuscript is under review, the London group published a similar study comparing the warhead chemistry of BTK degraders[17]. In this study, their reversible covalent BTK PROTAC RC-2[17] was less potent for BTK degradation compared with the irreversible covalent and reversible noncovalent counterparts. The main difference between our reversible covalent PROTAC and theirs is the dimethyl group at the γ-position of the cyanoacrylamide group in RC-1, reducing the reactivity of the Michael acceptor and leading to a mM $K_d$ when reacting with thiols. The cyanoacrylamide in London's reversible covalent PROTACs RC-2[17] is too reactive toward thiols, rendering the PROTAC trapped extracellularly by cysteine in the medium. Later, they showed that adding the dimethyl group to their initial

reversible covalent BTK PROTAC to form RC-3[17] indeed significantly increased its potency. There are still some discrepancies between their[17] and our studies. They showed that the irreversible covalent PROTAC can degrade both wild-type BTK and the C481S mutant, while the reversible covalent PROTAC can only degrade wild-type BTK not the C481S mutant. In our study, we found that RC-1 degrades both wild-type BTK and the C481S mutant with similar potency. Additionally, they showed that their reversible covalent and reversible noncovalent PROTACs have similar cellular concentrations based on a lysate-based LCMS measurement, contrary to our measurements of RC-1 and RNC-1 using the live-cell based NanoBRET assay. It should be noted that the PROTACs used in these two studies are not the same, particularly the linker lengths. The linker optimization we performed was focused on reversible covalent PROTACs. The reversible noncovalent BTK PROTACs may have a different optimal linker length from the reversible covalent counterparts. A side-by-side comparison of these BTK degraders is necessary to evaluate whether reversible covalent PROTACs are advantageous than the reversible noncovalent counterparts.

In summary, we develop a unique dual-functional BTK inhibitor and degrader and provide a general strategy to improve intracellular accumulation of PROTACs and other small molecules with poor cellular permeability. Through studying reversible covalent chemistry for PROTAC development, we serendipitously discover that the reversible covalent cyano-acrylamide moiety enhances intracellular accumulation of compounds. Our approach provides a potential solution to address intracellular accumulation issues, a major problem of PROTAC development, and can be generally applied to other small molecules with poor permeability. Unlike other PROTACs with low target occupancy due to poor permeability, RC-1 has high target occupancy and is effective as both an inhibitor and a degrader. This feature is highly important to ensure undegraded BTK protein would still be inhibited by RC-1. Otherwise, the remaining small portion of undegraded BTK could be sufficient to support the signaling cascades due to its catalytic nature. Additionally, molecular dynamics calculations and phase-separation-based ternary complex assays also support that RC-1 forms a stable ternary complex with BTK and CRBN. Lastly, RC-1 compares favorably with other reported BTK degraders in cell viability and target engagement assays and has a reasonable plasma half-life for in vivo applications. We hope this work can not only help to develop optimal BTK degraders for clinical applications but also provide another strategy to improve PROTAC efficacy.

## Methods

**Cell culture and treatment.** MOLM-14 cell line was obtained from Dr. Conneely at Baylor College of Medicine, and the Mino cells were purchased from the American Type Culture Collection (ATCC, Cat. No. CRL-3000). Both cell lines were cultured in RPMI 1640 medium (Thermo Fisher Scientific, Cat. No. MT10040CV) supplemented with 10% fetal bovine serum (GE Healthcare, Cat. No. SH30071.03) and 1% Pen/Strep (Thermo Fisher Scientific, Cat. No. 15140163). The wild-type and C481S BTK XLA cell lines were gifts from Dr. Woyach at The Ohio State University and were cultured in RPMI 1640 medium with 15% fetal bovine serum, 1% Pen/Strep, and 1.0 μg/ml puromycin (Thermo Fisher Scientific, Cat. No. A1113803). HEK 293T/17 cells were purchased from ATCC (Cat. No. CRL-11268) were maintained in DMEM (Thermo Fisher Scientific, Cat. No. MT10013CV) with 10% fetal bovine serum and 1% Pen-Strep. The B cell lymphoma cell line derived from Eu-Myc mice was a gift from Dr. Yulin Li at Houston Methodist Research Institute and cultured in RPMI 1640 medium supplemented with 15% (v/v) fetal bovine serum (FBS), 100 U/mL penicillin, 100 μg/mL streptomycin, and 0.05 mM beta-mercaptoethanol and puromycin (1 μg/mL). Cells were grown at 37 °C with 5% $CO_2$.

For dose-dependent BTK degradation studies, $1.0 \times 10^6$ MOLM-14 cells in 2 mL of RPMI 1640 complete media were incubated with indicated doses of BTK PROTAC compounds for 24 h, with control cells treated with 0.01% DMSO. XLA cells overexpressing wild-type BTK or mutant C481S BTK were plated at the same cell number and density as the above experiments and were treated with RC-1, IRC-1, or RNC-1 at the dose of 1.6, 8.0, 40, 200, and 1000 nM for 24 h. After

completion of treatment, all the cells were collected and processed for western blot analysis.

**Proteomics global profile of compound-treated cells.** For proteomic study of the cellular protein profile following BTK degradation, MOLM-14 cells were treated in duplicate with 200 nM of RC-1, RNC-1, IRC-1, or RC-1-Me for 24 h, and then the cells were collected and processed for proteomics analysis (Sanford Burnham Prebys Medical Discovery Institute, La Jolla, California).

For cell lysis and protein digestion, cells were lysed with vigorous shaking (20 Hz for 10 min at room temperature by Retsch MM301 instrument) in 0.5 mL of lysis buffer (8 M urea, 50 mM ammonium bicarbonate (ABC); plus Benzonase 24U/100 ml), and extracted proteins were centrifuged at $14,000 \times g$ for 10 min to remove cellular debris. Supernatant protein concentration was determined using a bicinchoninic acid (BCA) protein assay. Following quantification, the proteins were digested in solution. First, protein disulfide bonds were reduced with 5 mM tris(2-carboxyethyl)phosphine (TCEP) at 30 °C for 60 min, alkylated (carbamidomethylated) with 15 mM iodoacetamide (IAA) in the dark at room temperature for 30 min. Urea was then diluted to 1 M urea using 50 mM ammonium bicarbonate, and proteins were subjected to overnight digestion with mass spec grade Trypsin/Lys-C mix. Following digestion, samples were acidified with formic acid (FA) and subsequently desalted using Waters Sep-Pak C18. Cartridges were sequentially conditioned with 100% acetonitrile (ACN) and 0.1% FA, samples were then loaded, washed with 0.1% FA, and peptides eluted with 60% ACN, 0.1% FA. Finally, the organic solvent was removed in a SpeedVac concentrator prior to LC-MS/MS analysis.

Prior to TMT labeling, sample peptide amount was quantified using a NanoDrop[TM] spectrophotometer (ThermoFisher). Samples were labeled according to the table schematic shown below using our TMT protocol. Briefly, a normalized amount of peptides sample was resuspended in 200 mM HEPES pH 8, and then labeled with one of the TMT10-plex reagents for 1 h at room temperature. TMT labels/channels were as follows:

126/0: DMSO-1, 127 N/1: DMSO-2, 127 C/2: IRC1-1, 128 N/3: IRC1-2, 128 C/4: RC1-1, 129 N/5: RC1-2, 129 C/6: RNC1-1, 130 N/7: RNC1-2, 130 C/8: RC1-Me-1, and 131/9: RC1-Me-2.

Reactions were quenched with 5% hydroxylamine solution. The 10 samples were then combined, dried down in a SpeedVac system, resuspended in 0.1% FA, and desalted as described above. After desalting, the organic solvent was removed in a SpeedVac concentrator.

Dried samples were reconstituted in 20 mM ammonium formate pH ~10, and fractionated by high pH reverse phase using a Waters Acquity CSH C18 2.1-μm 1 × 150 mm column mounted on an M-Class Ultra Performance Liquid Chromatography (UPLC) system (Waters corp., Milford, MA). Peptides were separated in a 36-min linear gradient at a flow rate of 10 μl/min, consisting in: 1–5% in 1.4 min, 5–18% B in 3.5 min, 18–36% B in 20 min, 36–46% B in 2 min, 46–60% B in 5 min, and 60–70% B in 5 min ($A$ = 20 mM ammonium formate, pH 10; $B$ = 100% ACN; uv: 215-280 nm, 10 mm path length). A total of 36 fractions were collected and pooled in a non-contiguous manner into 18 total fractions: 1 + 19, 2 + 20, 3 + 21, 4 + 22, 5 + 23, 6 + 24, 7 + 25, 8 + 26, 9 + 27, 10 + 28, 11 + 29, 12 + 30, 13 + 31, 14 + 32, 15 + 33, 16 + 34, 17 + 35, and 18 + 36. Pooled fractions were dried to completeness in a SpeedVac concentrator.

Dried fractions were reconstituted with 2% ACN-0.1% FA and analyzed by LC-MS/MS using a nanoACQUITY system (Waters) coupled to an Orbitrap Fusion Lumos mass spectrometer (Thermo Fisher Scientific). Peptides were separated using an analytical C18 Acclaim PepMap column (75 μm × 250 mm, 2 μm particles; Thermo Scientific) using a 74-min gradient, at flow rate of 300 μl/min, consisting in: 1–5% B in 1 min, 5–18% B in 44 min, 18–27% B in 28 min, 27–38% B in 2 min, and 38–80% B in 2 min ($A$ = FA 0.1%; $B$ = 100% ACN: 0.1% FA). The mass spectrometer was operated in positive data-dependent acquisition mode. MS1 spectra were measured in the Orbitrap with a resolution of 60,000 (AGC target: 4e5; maximum injection time: 50 ms; mass range: from 350 to 1500 $m/z$). The instrument was set to run in top speed mode with 3 s cycles for the survey and the MS/MS scans. After a survey scan, tandem MS was performed in the Ion Routing Multipole HCD-Cell on the most abundant precursors by isolating them in the quadrupole (Isolation window: 0.7 $m/z$; charge state: +2–7; collision energy: 35%). Resulting fragments were detected in the Orbitrap at 50,000 resolution (First mass: 110 $m/z$; AGC target for MS/MS: 1e5; maximum injection time: 105 ms). The dynamic exclusion was set to 20 s with a 10-ppm mass tolerance around the precursor and its isotopes.

For data analysis, all mass spectra were analyzed with MaxQuant software version 1.5.5.1. MS/MS spectra were searched against the *Homo sapiens* Uniprot protein sequence database (version January 2018) and GPM cRAP sequences (commonly known protein contaminants). Reporter ion MS2 type was selected along with TMT 10plex option. Precursor mass tolerance was set to 20 ppm and 4.5 ppm for the first search where initial mass recalibration was completed and for the main search, respectively. Product ions were searched with a mass tolerance 0.5 Da. The maximum precursor ion charge state used for searching was 7. Carbamidomethylation of cysteines was searched as a fixed modification, while oxidation of methionines and acetylation of protein N-terminal were searched as variable modifications. Enzyme was set to trypsin in a specific mode and a maximum of two missed cleavages was allowed for searching.

The target-decoy-based false discovery rate (FDR) filter for spectrum and protein identification was set to 1%.

The corrected reporter intensities for each identified protein were used to quantify its relative abundance with different treatments (duplicates for each treatment condition). The statistical significance ($P$ value) was calculated using Student's $t$ test with two-tailed distribution and two-sample unequal variance (calculated using Microsoft Excel 365 and plotted using GraphPad Prism 7.0).

**Immunoblotting**. The collected cells and spleen tissues were lysed in RIPA lysis buffer (50 mM Tris-HCl, pH 7.4, 150 mM NaCl, 1% NP-40, 0.5% sodium deoxycholate, 0.1% SDS) supplemented with Halt™ Protease and Phosphatase Inhibitor Cocktail (Thermo Fisher Scientific, Cat. No. PI78440) immediately before use. Lysates were centrifuged at $15,000 \times g$ for 10 min at 4 °C and the supernatant was quantified for total protein concentration using the Pierce BCA Protein Assay (Thermo Fisher Scientific, Cat. No. 23225). Thirty micrograms of protein were loaded onto sodium dodecyl sulfate−polyacrylamide gel for electrophoresis (Bio-Rad) and transferred onto PVDF membranes (Millipore, Cat. No. IPVH00010). The membranes were probed with the specified primary antibodies at the dilution of 1:1000 (Cell Signaling Technology: anti-BTK, Cat. No. 8547; anti-CRBN, Cat. No. 71810; anti-p-BTK (Y223), Cat. No. 5082; anti-IKZF1, Cat. No. 14859; anti-IKZF3, Cat. No. 15103; anti-GAPDH, Cat. No. 2118; and anti-β-actin, Cat. No. 4970) overnight at 4 °C and the HRP-conjugated secondary antibodies (BIO-RAD, Cat. No. 1706515, 1:1000) for 1 h at room temperature. Imaging was performed using the ECL Prime chemiluminescent western blot detection reagents (GE Healthcare, Cat. No. RPN2232) by visualization of the blots to X-Ray film or with an Imager (Kindle Biosciences, Cat. No. D1001). All western blots were subsequently processed and quantified with Imager software. Protein level was normalized to β-actin or GAPDH loading controls.

**AlamarBlue cell viability assay**. MOLM-14 or Mino cells were harvested in the log phase of growth and re-plated into the wells of 96-well plates at the density of $6 \times 10^4$ cells per mL in 100 μL of complete RPMI-1640 culture medium. After overnight recovery, cells were exposed to serially diluted BTK PROTAC compounds (from 10,000 to 0.64 nM, 5-fold dilution) for 72 h, which was followed by adding of pre-warmed Resazurin sodium (Sigma, Cat. No. 199303) solution (1 mg mL$^{-1}$ in PBS) in an amount equal to 10% of the volume in the well. Four hours after incubation, fluorescence signals were collected with a BioTek SYNERGY H1 microplate reader at excitation/emission 544/590 nm from top with gain at 60. The IC$_{50}$ was calculated using GraphPad Prism software with a four-parameter dose-response curve fit.

**NanoBRET in-cell target engagement assay**. The target engagement assay for BTK or CRBN was from Promega (BTK kit, Cat. No. N2500; CRBN kit, Cat. No. CS1810C136) and was performed according to the manufacture's instruction with some modifications. Briefly, HEK 293/T17 cells (ATCC) were transiently transfected with BTK-NanoLuc® fusion vector or NanoLuc®-CRBN fusion vector (Promega) in 0.125 M CaCl$_2$ and 1x HBS. Forty-eight hours after transfection, the cells were resuspended in Opti-MEM medium (Life Technologies, Cat. No. 11058021) at the density of $2 \times 10^5$ cells per mL and were plated into 96-well plates (Corning, Cat. No. CLS3600). Cells were incubated with 1.0 μM (BTK) or 0.5 μM (CRBN) NanoBRET™ Tracer and serially diluted unlabeled BTK PROTAC compounds (from 10 to 0.0032 μM, 5-fold dilution) for 2 h at 37 °C with 5% CO$_2$ in an incubator. After 2 h incubation, the 3X Nano-Glo® Substrate with NanoLuc® extracellular inhibitor were added to cells and developed for 3 min at room temperature. BRET signals were collected using a BioTek SYNERGY H1 microplate reader equipped with a 450/80-nm BP filter for donor emission and a 610-nm LP filter for acceptor emission. BRET Ratio was calculated with the equation: [(Acceptor sample/Donor sample)–(Acceptor no tracer control/Donor no tracer control)] × 1000. The IC$_{50}$ of the compound against its BTK or CRBN tracer was calculated using GraphPad Prism software.

**In vitro kinase assay**. BTK kinase activity inhibition IC$_{50}$ was measured by PhosphoSens® Kinase Assay Kit (Assay Quant Technologies Inc., Cat.# CSKS-AQT0101K). This assay was performed in 384-well, white flat bottom polystyrene NBS microplates (Corning #3824) at room temperature. Active recombinant BTK was purchased from SignalChem (Cat.# B10-10H-10). All the drugs (3 warhead control and 4 PROTACs) were dissolved in DMSO (10 mM). Duplicate drug titrations (1000 nM, 200 nM, 40 nM, 8 nM, 1.6 nM, 0.32 nM 0.64 nM, 0.128 nM, and 0.0256 nM) were used to generate each IC$_{50}$. Typical final concentrations of each reaction component are as follows: 2.5 nM BTK, and 10 μM PhosphoSens® Substrate, 54 mM HEPES, pH 7.5, 1 mM ATP, 1.2 mM DTT, 0.012% Brij-35, 10 mM MgCl$_2$, 1% glycerol, and 0.2 mg mL$^{-1}$ BSA. Incubate the mixture at room temperature for 15 min and collect the fluorescence intensity readings (Ex 360 nm/Em 492 nm) for 60 min with 3 min interval in a BioTek Synergy H1 fluorescence microplate reader. To calculate the IC$_{50}$, subtract the background fluorescence determined with the "no kinase" control for each time point from the total signal to obtain corrected Relative Fluorescence Units (RFU) values. Plot the corrected RFU vs. Time for each inhibitor concentration and determine the initial reaction rates (slope of the linear portion) for each progress curve for each inhibitor

concentration. Then plot velocity (RFU corrected/minute) vs [inhibitor] and determine the IC$_{50}$ using a 4-parameter logistic fit.

**BTK-SPPIER assay**. All the plasmid constructs were created by standard molecular biology techniques and confirmed by exhaustively sequencing the cloned fragments. To create the E3 ubiquitin ligase CRBN-EGFP-HOTag3 fusion, full-length CRBN was first cloned into pcDNA3 containing EGFP. HOTag3 was then cloned into pcDNA3 E3 ligase-EGFP construct, resulting in pcDNA3 CRBN-EGFP-HOTag3. Similar procedures were carried out to produce pcDNA3 BTK$^{KD}$-EGFP-HOTag6.

The HEK293T/17 cells were passaged in Dulbecco's modified Eagle medium (DMEM) supplemented with 10% fetal bovine serum (FBS), nonessential amino acids, penicillin (100 units per mL), and streptomycin (100 μg mL$^{-1}$). All culture supplies were obtained from the UCSF Cell Culture Facility.

HEK293T/17 cells were transiently transfected with the plasmid using calcium phosphate transfection reagent or lipofectamine. Cells were grown in 35 mm glass bottom microwell (14 mm) dishes (MatTek Corporation). Transfection was performed when cells were cultured to ~50% confluence. For each transfection, 4.3 μg of plasmid DNA was mixed with 71 μL of 1X Hanks' Balanced Salts buffer (HBS) and 4.3 μL of 2.5 M CaCl$_2$. Cells were imaged 24 h after transient transfection. Time-lapse imaging was performed with the aid of an environmental control unit incubation chamber (InVivo Scientific), which was maintained at 37 °C and 5% CO$_2$. Fluorescence images were acquired with an exposure time of 50 ms for EGFP. Chemical reagents, including RC-1, IRC-1, and RNC-1, were carefully added to the cells in the incubation chamber when the time-lapse imaging started. Image acquisition was controlled by the NIS-Elements Ar Microscope Imaging Software (Nikon). Images were processed using NIS-Elements and ImageJ (NIH).

For analysis of the SPPIER signal, images were processed in imageJ. The sum of droplet pixel fluorescence intensity and cell pixel intensity was scored using the Analyze Particle function in imageJ.

**BTK binding assay**. The dissociation equilibrium constant $K_d$ was measured using the full-length BTK protein by Eurofins DiscoverX according to the following protocol: kinase-tagged T7 phage strains were prepared in an *E. coli* host derived from the BL21 strain. *E. coli* were grown to log-phase and infected with T7 phage and incubated with shaking at 32 °C until lysis. The lysates were centrifuged and filtered to remove cell debris. The remaining kinases were produced in HEK-293 cells and subsequently tagged with DNA for qPCR detection. Streptavidin-coated magnetic beads were treated with biotinylated small molecule ligands for 30 min at room temperature to generate affinity resins for kinase assays. The liganded beads were blocked with excess biotin and washed with blocking buffer (SeaBlock (Pierce), 1% BSA, 0.05% Tween 20, 1 mM DTT) to remove unbound ligand and to reduce non-specific binding. Binding reactions were assembled by combining kinases, liganded affinity beads, and test compounds in 1x binding buffer (20% SeaBlock, 0.17x PBS, 0.05% Tween 20, 6 mM DTT). Test compounds were prepared as 111X stocks in 100% DMSO. $K_d$ values were determined using an 11-point 3-fold compound dilution series with three DMSO control points. All compounds for $K_d$ measurements are distributed by acoustic transfer (non-contact dispensing) in 100% DMSO. The compounds were diluted directly into the assays such that the final concentration of DMSO was 0.9%. All reactions performed in polypropylene 384-well plate. Each was a final volume of 0.02 mL. The assay plates were incubated at room temperature with shaking for 1 h and the affinity beads were washed with wash buffer (1x PBS, 0.05% Tween 20). The beads were then resuspended in elution buffer (1x PBS, 0.05% Tween 20, 0.5 μM non-biotinylated affinity ligand) and incubated at room temperature with shaking for 30 min. The kinase concentration in the eluates was measured by qPCR. For kinase-binding constant determination, an 11-point 3-fold serial dilution of each test compound was prepared in 100% DMSO at 100x final test concentration and subsequently diluted to 1x in the assay (final DMSO concentration = 1%). Most $K_d$ values were determined using a compound top concentration = 30,000 nM. If the initial $K_d$ determined was <0.5 nM (the lowest concentration tested), the measurement was repeated with a serial dilution starting at a lower top concentration. A $K_d$ values reported as 40,000 nM indicates that the $K_d$ was determined to be >30,000 nM. Binding constants ($K_d$ values) were calculated with a standard dose-response curve using the Hill equation:

$$\text{Response} = \text{Background} + \frac{\text{Signal} - \text{Background}}{1 + \left( K_d^{\text{Hill Slope}} / \text{Dose}^{\text{Hill Slope}} \right)}$$

The Hill Slope was set to −1. Curves were fitted using a non-linear least square fit with the Levenberg-Marquardt algorithm.

**CRBN binding assay**. The $K_d$ values of compounds to CRBN were measured based on fluorescence quenching after compounds binding to CRBN. This assay was performed in 384-well, black plate (Greiner Bio-One #784076) at room temperature. All the compounds were dissolved in DMSO (10 mM) as stock solutions. The final drug concentration is 100 nM. Triplicate protein titrations (23.50 11.75, 5.88, 2.94, 1.47, and 0.73 μM) were used to generate each $K_d$. Incubate the mixture of drug and protein at room temperature for 15 min. Fluorescence signals were

collected with a BioTek SYNERGY H1 microplate reader at excitation/emission 400/485 nm from top with gain at 130. Fluorescence intensity values for control wells (assay buffer only) were subtracted from each data point prior to analysis. The $K_d$ was calculated using GraphPad Prism with the following equation curve fit.

$$I = (I_0 - I_f) \times \frac{K_d}{[P_0] + K_d} + I_f$$

**Quantification of intracellular drug concentration by LC-MS.** MOLM-14 cells were seeded at a density of $0.5 \times 10^6$ per mL in 2 mL medium. After overnight recovery, the cells were treated with either DMSO or 200 nM of compound for 2 h and collected into 2 mL centrifuge tubes. The samples were immediately centrifuged at $12,000 \times g$ for 15 min at 4 °C, then removing the supernatant. In all, 400 μL of NP-40 lysis buffer was added to the cell pellet and mixed thoroughly. The mixtures were rocked for 1 h at 4 °C and centrifuged at $12,000 \times g$ for 15 min at 4 °C. Then the supernatant was transferred to new EP tubes. In total, 20 μL supernatant was extracted with 180 μL of acetonitrile (containing internal standard: 250 nM of RNC-Ctrl) and centrifuged at the same conditions described above. Take 75 μL of supernatant for liquid chromatography-mass spectrometry analysis. The standard curve was established by serially diluted drug in cell lysates (from 5 μM to 0.0195 μM, 2-fold dilution) to calibrate all the results.

*For LC-MS condition*: the supernatant was analyzed by LC-MS (Agilent Technologies 1260 Infinity) with a Eclips Plus C18 column (4.6 μm × 100 mm, 3.5 μm particles; Agilent) using a 7-min gradient, at flow rate of 700 μl/min, consisting in: 20–100% B in 3 min, 100% B in 4 min ($A = 100\%$ water: 0.1% FA; $B = 100\%$ ACN: 0.1% FA). The mass spectrometer was operated in positive SIM ion mode (fragmentor 120 V; cycle time 0.6 s/cycle; Agilent Technologies 6130 Quadrupole LC/MS).

**Permeability determination using lipid-PAMPA method.** The permeability of compounds was measured using Lipid-PAMPA method by Pharmaron. Preparation of stock solutions: testosterone and methotrexate were used as control compounds in this assay. The stock solutions of positive controls were prepared in DMSO or acetonitrile at the concentration of 10 mM. Prepare a stock solution of compounds in DMSO at the concentration of 10 mM, and further dilute with PBS (pH 7.4). The final concentration of the test compound is 10 μM.

*Assay procedures*: prepare a 1.8% solution (w/v) of lecithin in dodecane, and sonicate the mixture to ensure a complete dissolution. Carefully pipette 5 μL of the lecithin/dodecane mixture into each acceptor plate well (top compartment), avoiding pipette tip contact with the membrane. Immediately after the application of the artificial membrane (within 10 min), add 300 μL of PBS (pH 7.4) solution to each well of the acceptor plate. Add 300 μL of drug-containing solutions to each well of the donor plate (bottom compartment) in triplicate. Slowly and carefully place the acceptor plate into the donor plate, making sure the underside of the membrane is in contact with the drug-containing solutions in all wells. Replace the plate lid and incubate at 25 °C, 60 rpm for 16 h. After incubation, aliquots of 50 μL from each well of acceptor and donor plate are transferred into a 96-well plate. Add 450 μL of methanol (containing IS: 100 nM Alprazolam, 200 nM Caffeine, 100 nM Tolbutamide) into each well. Cover with plate lid. Vortex at 750 rpm for 100 s. Samples were centrifuged at $3220 \times g$ for 20 min. Determine the compound concentrations by LC/MS/MS. The effective permeability (Pe), in units of centimeter per second, can be calculated using the following equation:

$$\log P_e = \log \left\{ C \times \left[ -L_n \left( 1 - \frac{[\text{drug}]_{\text{acceptor}}}{[\text{drug}]_{\text{equilibrium}}} \right) \right] \right\}$$

**Relative intracellular accumulation coefficient ($K'_{P,D}$).** The species in a Nano-Luc based target engagement assay include the target protein nanoLuc fusion (designated as P), the tracer (designated as T), the tracer bound target protein nanoLuc fusion (designated as PT), the drug (designated as D), and the drug bound target protein nanoLuc fusion (designated as PD). In this simplified model, we do not consider the binding of tracer T and drug D to intracellular proteins or biomolecules other than the target protein nanoLuc fusion P.

For the tracer T and the target protein nanoLuc fusion P interactions,

$$[P] + [T] \rightleftharpoons [PT] \tag{1}$$

$$K_{d,T} = \frac{[P][T]}{[PT]} \tag{2}$$

$$C_T = [T] + [PT] \tag{3}$$

$K_{d,T}$ is the dissociation equilibrium constant for the binding between the target protein nanoLuc fusion and the tracer; [P] is the equilibrium concentration of the target protein nanoLuc fusion in cells; [T] is the equilibrium concentration of the free tracer T in cells; [PT] is the equilibrium concentration of the tracer bound form of the target protein nanoLuc fusion in cells; and $C_T$ is the total concentration of tracer T in cells.

For the drug D and the target protein nanoLuc fusion P interactions,

$$[P] + [D] \rightleftharpoons [PD] \tag{4}$$

$$K_{d,D} = \frac{[P][D]}{[PD]} \tag{5}$$

$$C_{D,in} = [D] + [PD] \tag{6}$$

$K_{d,D}$ is the dissociation equilibrium constant for the binding between the target protein nanoLuc fusion and the drug; [P] is the equilibrium concentration of the target protein nanoLuc fusion in cells; [D] is the equilibrium concentration of the free drug D in cells; [DT] is the equilibrium concentration of the drug bound form of the target protein nanoLuc fusion in cells; and $C_{D,in}$ is the total concentration of drug D in cells.

Based on the fact that the target protein nanoLuc fusion P is either unbound or bound to T or D, we can deduce that

$$C_P = [P] + [PD] + [PT] \tag{7}$$

$C_P$ is the total concentration of the target protein nanoLuc fusion P in cells.

Without the addition of drug D, the tracer bound nanoLuc fusion P (PT) accounts for 50% of all the nanoLuc fusion P and corresponds to the concentration ½ $C_P$ (please note: promega recommends ~50% of all the nanoLuc fusion P bound with tracer T without addition of free drug competitor). When 50% of target engagement is reached with the addition of drug D to the medium, drug D competes off the tracer to leave only 25% of PT (that is 50% of 50% PT), we can deduce that

$$[PT] = \frac{1}{4} C_P \tag{8}$$

Based on Equation (3), $[T] = C_T - [PT] = C_T - \frac{1}{4} C_P \tag{9}$

Based on Equation (2), $[P] = K_{d,T} \frac{[PT]}{[T]} = K_{d,T} \frac{C_P}{4C_T - C_P} \tag{10}$

Based on Equation (7), $[PD] = C_P - [P] - [PT] = \frac{3}{4} C_P - K_{d,T} \frac{C_P}{4C_T - C_P} \tag{11}$

Based on Equation (5), $[D] = K_{d,D} \frac{[PD]}{[P]} \tag{12}$

The intracellular accumulation coefficient for drug D ($K_{P,D}$) can be defined as

$$K_{P,D} = \frac{C_{D,in}}{C_{D,ex}} \tag{13}$$

$C_{D,in}$ is the total intracellular concentration of drug D; $C_{D,ex}$ is the total extracellular concentration of drug D.

When 50% of target engagement is reached with the addition of drug D to the medium (i.e. $C_{D,ex} = \text{IC}_{50,TE}$ assuming intracellular drug accumulation does not significantly change the extracellular drug concentration), we can reasonably assume $[D] \gg [PD]$ due to the limited amount of the target protein nanoLuc fusion P in cells. Because $C_{D,in} = [D] + [PD]$, then we can approximate $C_{D,in} = [D]$. Therefore,

$$K_{P,D} = \frac{C_{D,in}}{C_{D,ex}} = \frac{[D]}{\text{IC}_{50,TE}} = \frac{K_{d,D}}{\text{IC}_{50,TE}} \frac{[PD]}{[P]} \tag{14}$$

Under the same assay conditions, $C_P$, $C_T$, and $K_{d,T}$ are constants. Therefore, based on Eqs. 8 and 9, [PD]/[P] is a constant, which can be defined as $A$. So, we can deduce

$$K_{P,D} = A \frac{K_{d,D}}{\text{IC}_{50,TE}} \tag{15}$$

We further define the relative intracellular accumulation coefficient for drug D ($K'_{P,D}$) as

$$K'_{P,D} = \frac{K_{P,D}}{A} = \frac{K_{d,D}}{\text{IC}_{50,TE}} \tag{16}$$

Under the same assay conditions, a greater $K'_{P,D}$ value for a drug reflects its higher propensity to accumulate inside cells.

**Animals.** Female ICR mice (weighing 22–28 g) were obtained from the Center for Comparative Medicine of Baylor College of Medicine. Mice were housed 2–4 per cage in an American Animal Association Laboratory Animal Care accredited facility and maintained under standard conditions of temperature (22 ± 2 °C), relative humidity (50%) and light and dark cycle (12/12 h), and had access to food and water ad libitum. Mice were allowed to acclimate to their environment for one week before experiment. We have complied with all relevant ethical regulations for animal testing and research. All the animal experiments were approved by the Institutional Animal Care and Use Committee (IACUC) at Baylor College of Medicine.

**Pharmacokinetic and pharmacodynamic studies of RC-1.** The pharmacokinetic profile of the BTK PROTAC RC-1 was evaluated in female ICR mice (weighing 22-28 g, $n = 3$). RC-1 was formulated in a non-aqueous solvent (30% PEG-400, 5% Tween 80, and 5% DMSO in deionized water) and was administered in a single IP injection (20 mg kg$^{-1}$). Blood samples (25 μL) were withdrawn from the tail vein at the time points of 10 min, 30 min, 1 h, 2 h, 4 h, 8 h, and up to 24 h after dosing. The blood samples were collected into 1.5 mL centrifuge tubes coated with 0.5 M EDTA and were immediately centrifuged at 12,000 × g at 4 °C for 15 min. The resultant plasma was extracted with 9 vol of acetonitrile (containing internal standard: 250 nM of RNC-Ctrl) and then were centrifuged at the same conditions described above. The supernatant was stored at −80 °C for liquid chromatography-mass spectrometry analysis. The standard curve was established by serially diluted RC-1 in plasma collected from naïve mice (from 5.0 to 0.003 μg mL$^{-1}$, 4-fold dilution). The PK parameters were calculated with the Microsoft Excel PK Solver.

To investigate the protein degradation effect of RC-1 in vivo, mice were subjected to single RC-1 administration at the dose of 50 or 100 mg kg$^{-1}$ (i.p.) or seven once-daily RC-1 administration at the dose of 20 mg kg$^{-1}$ (i.p.) ($n = 2$-3). Twenty-four hours after the one-time RC-1 injection or 24 h after the 7th injection, mice were anesthetized with isoflurane and the whole spleen was dissected and stored at −80 °C for western blot analysis of protein level.

**Molecular dynamics modeling.** Construction of initial model for ternary complexes of {BTK-Ligands-CRBN}: To construct the ternary complexes of {BTK-Ligands-CRBN}, the binary complexes of {BTK-CRBN} was firstly modeled. The structure files of BTK-ibrutinib (PDB code: 5P9J) and that of CRBN-lenalidomide (PDB code: 4TZ4) were downloaded from RCSB (http://www.rcsb.org/)[51]. Multiple residues at the entrance of the protein BTK (L408, N484 and K558) and CRBN (I371, H353 and E377) were randomly selected as the indication of active binding site, and constrained protein-protein docking simulations were performed by using ZDOCK[52]. Three candidate binding conformations were reported and downloaded from the ZDOCK server. Since the protein-protein binding interface of the second conformation is in between the entrance of active sites of BTK and CRBN, this predicted BTK-CRBN complex was selected for ligand docking. To select a proper ligand conformation for our follow-up MD simulation, the low-energetic molecular conformations of the linker designed for PROTACs were sampled by our in-house small molecular conformational sampling method Cyndi[53,54]. The linker structure whose length can fit without steric clashes in between BTK and CRBN of the chosen protein complex was selected. Finally, the linker was artificially attached to the ibrutinib and lenalidomide for constructing the initial ternary complexes. A total of three complexes were constructed for {BTK-RNC-1-CRBN}, {BTK-IRC-1-CRBN}, and {BTK-RC-1-CRBN}, respectively.

Refining ternary complexes of {BTK-Ligands-CRBN} by molecular dynamic simulations: to further refine the structures of our selected {BTK-Ligands-CRBN} complexes and to assess their stability under a physically motivated force field, the initial complex structure was solvated in an all-atom explicit-solvent environment for molecular dynamics simulation. Appropriate number of counter ions were added to neutralize the system. A cubic box was used with periodic boundary condition to ensure no artificial contacts between solute and its images, that results in a total of ~270 K atoms for each of the three simulation systems. Energy minimization was performed using steepest decent method until the maximum force is smaller than 1000.0 kJ mol$^{-1}$nm$^{-1}$. The system was further equilibrated for 1 ns under NVT and NPT ensemble, respectively, before continuing with a production run of more than 200 ns. During equilibration run, the protein and ligand position are constrained by a harmonic potential. For both the equilibration and production runs, the temperature was set to 300 K using the V-rescale thermostat[55] separately for protein-ligand complex and solution. The coupling constant of the external thermal bath was set to 0.1 ps. Also, the pressure of the system was set to 1 atm (Parrinello-Rahman coupling[56]). Since one covalent bond is formed in between the residue C481 of BTK and the ligands RC-1 and IRC-1, a harmonic potential with the spring constant of 167360 kJ mol$^{-1}$nm$^{-2}$ was added in between the corresponding sulfur and carbon atoms to mimic this additional constraint. All simulations were performed using the CHARMM36 force field[57] and with molecular simulation package GROMACS-2019.2[58].

**Reporting summary.** Further information on research design is available in the Nature Research Reporting Summary linked to this article.

## Data availability

All mass spectrometry raw files been deposited in the MassIVE repository housed at UCSD (https://massive.ucsd.edu/) with the accession number MSV000084902 (https://doi.org/10.25345/C5H11W). All other data are available from the corresponding authors on reasonable request. Source data are provided with this paper.

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

## Acknowledgements

The research was supported in part by National Institute of Health (R01-GM115622 and R01-CA250503 to J.W.), NMR and Drug Metabolism Core Facility Support Award from the Cancer Prevention Research Institute of Texas (RP160805), Protein and Monoclonal Antibody Production Core (P30 Cancer Center Support Grant NCI-CA125123), Baylor College of Medicine (seed funding to J.W.) and Howard Hughes Medical Institute (to M.C.W.). Work at the Center for Theoretical Biological Physics was sponsored by the NSF (Grants PHY-2019745 and CHE-1614101) and by the Welch Foundation (Grant C-1792). J.N.O. is a Cancer Prevention and Research Institute of Texas (CPRIT) Scholar in Cancer Research. We also thank Dr. Alex Rosa Campos at the Proteomics Core (supported by NIH P30 CA030199) from Sanford Burnham Prebys for his help with the proteomics experiment, Dr. Matthew Calabrese at Pfizer for sharing the plasmid sequences of truncated CRBN, Dr. Yulin Li at Houston Methodist Research Institute for providing the B cell lymphoma cell line derived from Eu-Myc mice, and Drs. Danette Daniels, Matt Robers, and Jim Vasta at Promega for their advice on the NanoBRET assays. C.-I.C. and X.S. designed and performed the BTK-SPPIER assay.

## Author contributions

W.G, X.Q, and J.W. conceived the idea and design of the study. W.G, X.Q., X.Y., Y.L., C.C, F.B., X.L, D.L, L.W., J.C., and L.H.S. performed the experiments. F.L., M.C.W., X.S., J.N.O., J.A.W., and M.L.W. provided reagents. W.G, X.Q., K.J.N., M.L.W., and J.W. analyzed the data and wrote the manuscript.

## Competing interests

J.W., W.G., X.Q., M.L.W., K.N., and Y.L. are the co-inventors of a patent related to this work. J.W. is the co-founder of CoActigon Inc. and Chemical Biology Probes LLC. X.S. and C.-I.C. have filed a patent application covering the SPPIER assay. M.L.W. receives research grants from Janssen, Pharmacyclics, AstraZeneca, Acerta Pharma, Celgene, Juno Therapeutics, BeiGene, Kite Pharma, Loxo Oncology, VelosBio, BioInvent, Aviara, Verastem, and InnoCare, consults and/or serves on the advisory board for Consulting/Advisory Board for Loxo Oncology, Janssen, Pharmacyclics, BioInvent, Celgene, Juno Therapeutics, Pulse Biosciences, MoreHealth, Guidepoint Global, Kite Pharma, AstraZeneca, Acerta Pharma, and InnoCare, received honoraria from Janssen, Acerta Pharma, OMI, Physicians Education Resources (PER), Oncology News, and Targeted Oncology, and owns stocks from MoreHealth. J.A.W. consults for Janssen, Pharmacyclics, Abbvie, AstraZeneca, and Arqule, receives research funding from Loxo Oncology and Abbvie and clinical trial funding from Pharmacyclics, Janssen, Verastem, Karyopharm, and Morphosys.
