## [Peer Review File · Nature Communications]

Reviewers' comments:

Reviewer #1 (Remarks to the Author):

This is an interesting study of the effect of reversible covalent Protac ligands on protein degradation. The authors provide compelling data for the effectiveness of their RC-1 ligand in producing BTK degradation. I cannot comment on the underlying chemistry but the biological experiments have been carefully performed and generally well analysed (but see below).

I do, however, have a few issues with some of the target engagement calculations and the determination of IC₅₀ values using the NanoBRET ligand binding approach. According to Figure 4 the IC₅₀ values cited in the Tables have been calculated from the concentration that reduces the NanoBRET signal by 50% - which is not correct. The IC₅₀ value should be calculated from the specific component of binding after subtraction of non-specific binding. In Fig 4b it is clear that the inhibition curve obtained with RC-1 levels off significantly above zero binding. This non-displaceable binding is likely to represent non-specific binding. The IC₅₀ will therefore be at lower concentrations than currently calculated. The same is true for Fig 4c and Fig 4d. Some information should be provided on the occupancy of the fixed concentration of fluorescent ligand used in the assay.

Target Engagement IC₅₀ is not “the compound concentration required to achieve 50% BTK target engagement measured using a NanoBRET BTK in-cell target engagement assay” as described in the Table legends. It is the concentration of compound required to inhibit binding of the fixed concentration of fluorescent probe. The IC₅₀ will very much depend on the concentration of probe used in the experiments. A much clear explanation is required on this.

I also do not fully understand what is meant by ‘normalised target engagement’ in Table 1 and 2. Occupancy (target engagement) is normally represented by $\frac{[\text{ligand}]}{([\text{Ligand}] + K_d)}$. What do these normalised target engagement values represent in the Tables?

I struggled to understand the fluorophore phase separation-based assay for imaging ternary complex formation. It looked to me to be a general measure of protein aggregation with the complexes being quite large (as judged by their size with respect to the scale bar). I failed to understand how this was a measure of ternary complex formation. Surely the spatial resolution is too low?

Reviewer #2 (Remarks to the Author):

In this manuscript, Guo et al. developed RC-1 as the first reversible covalent BTK PROTAC, which has high target occupancy and is effective both as an inhibitor and a degrader. They provided biochemical and cellular data to support their conclusion that RC-1 induces mechanism-based BTK degradation against both wild-type BTK and C481S mutant form.

The authors compared RC-1 with reversible noncovalent degrader (RNC-1) and irreversible covalent degrader (IRC-1) and found RC-1 shows better activity in cells and suggested that reversible covalent degrader could be a good strategy for the development of PROTACs. In addition, they demonstrated that the enhanced cellular permeability is responsible for better cellular potencies for RC-1 than RNC-1 and IRC-1. Their study suggests that “introducing a cyano-acrylamide moiety can be a general strategy to enhance cellular permeability for PROTACs.” To test this idea further, they have designed FLT3 degraders and obtained similar results. Finally, they tested RC-1 in vivo, shows BTK degradation in spleen, which suggests reversible covalent BTK degraders may be useful for in vivo studies.

Overall, this study was well done and is significant for the field of PROTACs. It is recommended for publication in Nature Communications with the following revisions.

1. Since the most attractive point of the manuscript is “introducing a cyano-acrylamide moiety can be a general strategy to enhance cellular permeability for PROTACs”, it is more convincing to provide direct evidence about the improved cellular permeability for RC-1 over other compounds using the Caco-2 cell-permeability assay or other similar assays.
2. It will be useful to make another reversible noncovalent compound as reference, which is directly reduce the double bond of RC-1. Compare its cell permeability with RC-1 may give us more information, such as whether the enhanced permeability is caused by adding lipophilic groups (cyano and di-methyl) or because of the reversible covalent ability.
3. Have the authors compared the BTK degradation between Crews’s compound (such as MT802) and RC-1 in Momi-14 cell line or other cell lines? It will be useful to compare these two compounds, since MT802 is an optimal reversible noncovalent degrader.
4. In the in vivo PD study, the drug was dosed for 7 days. The PD effect is not as good what has been observed in vitro. Since degradation of BTK is already profound within 24 hours in cells, what was the reason for the 7-day dosing and for the relative less profound degradation of BTK?

Reviewer #3 (Remarks to the Author):

This is a well-done and timely study on the effect of semi-reversible BTK PROTACs and showing that these molecules are able to degrade BTK in cells. They perform followup proteomic studies and also compare their PROTACs to other BTK PROTACs that have been previously synthesized. This study complements recent cysteine-targeting semi-reversible BTK PROTAC papers from Nir London's lab and Pfizer that have been recently posted on ChemRxiv and BioRxiv. This study should be published as is.

Reviewer #1 (Remarks to the Author):

This is an interesting study of the effect of reversible covalent Protac ligands on protein degradation. The authors provide compelling data for the effectiveness of their RC-1 ligand in producing BTK degradation. I cannot comment on the underlying chemistry but the biological experiments have been carefully performed and generally well analyzed (but see below).

1. I do, however, have a few issues with some of the target engagement calculations and the determination of IC₅₀ values using the NanoBRET ligand binding approach. According to Figure 4 the IC₅₀ values cited in the Tables have been calculated from the concentration that reduces the NanoBRET signal by 50% - which is not correct. The IC₅₀ value should be calculated from the specific component of binding after subtraction of non-specific binding. In Fig 4b it is clear that the inhibition curve obtained with RC-1 levels off significantly above zero binding. This non-displaceable binding is likely to represent non-specific binding. The IC₅₀ will therefore be at lower concentrations than currently calculated. The same is true for Fig 4c and Fig 4d.

We thank the reviewer for pointing this out and agree that our original calculations for target engagement IC₅₀ values were incorrect. It is true that the non-displaceable binding at high concentrations may be due to non-specific binding to the target protein nanoLuc fusion. Following Promega's data analysis in *Nat Commun* **6**, 10091 (2015). <https://doi.org/10.1038/ncomms10091>. "competitive displacement data were then graphed with GraphPad Prism software using a three-parameter curve fit with the following equation. Normalized data were generated by assigning 100% to the theoretical maximum of the three-parameter curve fit and 0% for the theoretical minimum value of the three-parameter curve fit"

$$Y = Bottom + (Top - Bottom) / (1 + 10^{(X - LogIC_{50})})$$

We have subtracted the background and recalculated all the target engagement IC₅₀ values in the revised manuscript.

2. Some information should be provided on the occupancy of the fixed concentration of fluorescent ligand used in the assay.

We thank the reviewer for pointing this out. The fixed concentrations of the fluorescent ligands used in the target engagement assays were mentioned in the supporting information. The concentrations of the fluorescent ligands are 1 μM and 0.5 μM for the BTK and CRBN target engagement assays, respectively. We included this information in the figure legend, such as "Figure 4 (B) CRBN in-cell target engagement assay. HEK-293 cells were transiently transfected with plasmids expressing a fusion protein of CRBN and nano-luciferase (nLuc) for 24 h and then the cells were treated with a CRBN tracer (**0.5 μM**), which binds to CRBN to induce bioluminescence resonance energy transfer signals".

3. Target Engagement IC₅₀ is not "the compound concentration required to achieve 50% BTK target engagement measured using a NanoBRET BTK in-cell target engagement assay" as described in the Table legends. It is the concentration of compound required to inhibit binding of the fixed concentration of fluorescent probe. The IC₅₀ will very much depend on the concentration of probe used in the experiments. A much clear explanation is required on this.

We thank the reviewer for pointing this out and agree that our original calculations for target engagement IC₅₀ values were incorrect. We have corrected it as "Target Engagement IC₅₀ value is the concentration of an unlabeled compound that results in a half-maximal inhibition binding of the fluorescent tracer" in the revised manuscript.

It is true that “the IC_{50} will very much depend on the concentration of probe used”. Promega have checked the effect of different concentrations of tracers. As the Promega’s results shown below, generally a higher tracer concentration leads to a higher Target Engagement IC_{50} value. We performed these target engagement assays using the Promega’s kits. Considering the assay window, we picked up Promega’s recommended tracer concentrations in our experiment, which are 1 μM and 0.5 μM for BTK and CRBN tracers, respectively. We have added these details in the Method section. It should be noted that under the same assay conditions, the target engagement IC_{50} values are comparable for different drugs and inversely proportional to the tendency for drugs to accumulate inside cells.

4. I also do not fully understand what is meant by ‘normalised target engagement’ in Table 1 and 2. Occupancy (target engagement) is normally represented by $[\text{ligand}]/([\text{Ligand}] + K_d)$. What do these normalized target engagement values represent in the Tables?

Sorry for the confusion. To better understand the target engagement data, we introduced the concept of intracellular accumulation coefficient for drug D ($K_{P,D}$) defined as $K_{P,D} = \frac{C_{D,in}}{C_{D,ex}}$, where $C_{D,in}$ and $C_{D,ex}$ are the total intracellular and extracellular concentrations of drug D, respectively. Based on our analysis (see SI for details), we can deduce the relative intracellular accumulation coefficient for drug D ($K'_{P,D}$) as $K'_{P,D} = \frac{K_{d,D}}{IC_{50,TE}}$, where $K_{d,D}$ is the dissociation equilibrium constant for the binding between the target protein nanoLuc fusion and the drug and $IC_{50,TE}$ is the IC_{50} value to reach 50% of target engagement in the NanoLuc assay. Under the same assay conditions, a greater $K'_{P,D}$ value for a drug reflects its higher tendency to accumulate inside cells. If all the compounds tested in the NanoLuc assay have the same binding affinities to the target protein (i.e. the same $K_{d,D}$ value, so no need for normalization), $K'_{P,D}$ will be inversely proportional to the $IC_{50,TE}$ values.

5. I struggled to understand the fluorophore phase separation-based assay for imaging ternary complex formation. It looked to me to be a general measure of protein aggregation with the complexes being quite large (as judged by their size with respect to the scale bar). I failed to understand how this was a measure of ternary complex formation. Surely the spatial resolution is too low?

Sorry for the confusion. The assay for “Fluorophore Phase Transition-Based Imaging of Ternary Complex Formation” is based on our collaborator Dr. Xiaokun Shu’s recent study published in Analytical Chemistry 2018 (ref 30). We rewrote this paragraph with additional background information. It should be noted that this assay is NOT a direct measurement of a single ternary complex formation, which is too small to be imaged using a confocal microscope. Instead many ternary complexes are crosslinked together to form condensates in cells, which then can be conveniently imaged with a microscope. The sizes of the EGFP droplets are not correlated with the sizes of the ternary complex. We hope it clarifies the confusion.

Fluorophore Phase Transition-Based Imaging of Ternary Complex Formation in Live Cells

To visualize small molecule-induced protein-protein interactions (PPIs), we recently applied fluorophore phase transition-based principle and designed a PPI assay named SPPIER (separation of phases-based protein interaction reporter)³⁰. A SPPIER protein design includes three domains, a protein-of-interest, an enhanced GFP (EGFP) and a homo-oligomeric tag (HOTag). Upon small molecule-induced PPI between two proteins-of-interest, multivalent PPIs from HOTags drive EGFP phase separation, forming brightly fluorescent droplets³⁰. Here, to detect PROTAC-induced PPI between BTK and CRBN, we engineered the kinase domain of BTK (amino acid residues 382 – 659, referred to as BTK^{KD}) into SPPIER to produce a BTK^{KD}-EGFP-HOTag6 construct, which forms tetramers when expressing in cells (**Figure 5A**). The previously reported CRBN-EGFP-HOTag3 fusion construct³⁰, which forms hexamers in cells, was used as the E3 ligase SPPIER (**Figure 5A**). If PROTACs can induce {BTK-PROTAC-CRBN} ternary complex formation in cells, they will crosslink the BTK^{KD}-EGFP-HOTag6 tetramers and the CRBN-EGFP-HOTag3 hexamers to produce EGFP phase separation, which can be conveniently visualized with a fluorescence microscope. This assay is named as BTK-SPPIER. HEK293 T/17 cells were transiently transfected with both constructs. Twenty-four hours after transfection, the cells were incubated with 10 μ M of RC-1, IRC-1 or RNC-1. Live cell fluorescence imaging revealed that RC-1, but not IRC-1 nor RNC-1, induced appreciable green fluorescent droplets (**Figure 5B**). This imaging result indicated that RC-1 is more efficient to induce {BTK-PROTAC-CRBN} ternary complex formation in living cells than RNC-1 and IRC-1 under the same experimental conditions. It should be noted that the concentration needed for RC-1 to induce appreciable droplet formation in this assay is much higher than its DC₅₀ in MOLM-14 cells potentially due to the high overexpression of the target proteins and the sensitivities of the assay.

Reviewer #2 (Remarks to the Author):

In this manuscript, Guo et al. developed RC-1 as the first reversible covalent BTK PROTAC, which has high target occupancy and is effective both as an inhibitor and a degrader. They provided biochemical and cellular data to support their conclusion that RC-1 induces mechanism-based BTK degradation against both wild-type BTK and C481S mutant form. The authors compared RC-1 with reversible noncovalent degrader (RNC-1) and irreversible covalent degrader (IRC-1) and found RC-1 shows better activity in cells and suggested that reversible covalent degrader could be a good strategy for the development of PROTACs. In addition, they demonstrated that the enhanced cellular permeability is responsible for better cellular potencies for RC-1 than RNC-1 and IRC-1. Their study suggests that “introducing a cyano-acrylamide moiety can be a general strategy to enhance cellular permeability for PROTACs.” To test this idea further, they have designed FLT3 degraders and obtained similar results. Finally, they tested RC-1 in vivo, shows BTK degradation in spleen, which suggests reversible covalent BTK degraders may be useful for in vivo studies.

Overall, this study was well done and is significant for the field of PROTACs. It is recommended for publication in Nature Communications with the following revisions.

1. Since the most attractive point of the manuscript is “introducing a cyano-acrylamide moiety can be a general strategy to enhance cellular permeability for PROTACs”, it is more convincing to provide direct evidence about the improved cellular permeability for RC-1 over other compounds using the Caco-2 cell-permeability assay or other similar assays.

We thank the reviewer for this suggestion. In fact, the permeabilities of RC-1, IRC-1 and RNC-1 were measured using the lipid-PAMPA method (performed by Pharmaron Inc.). None of the compounds were detected on the receptor side, indicating extremely poor physical permeabilities. Additionally, the total recovery rates for all the three compounds were only in the range of 20-30%, suggesting that these compounds may stick to the PAMPA lipids and/or the plasticware used in this assay (**Table S3**). Therefore, we concluded that the lipid-PAMPA method is not an ideal assay to evaluate the permeabilities of our PROTACs.

Based on our studies, we prefer the term “enhanced intracellular accumulation” instead of “enhanced permeability” to describe the unique feature of RC-1. Permeability characterizes the potential for a compound to

transverse the plasma membrane of cells, which can be energy independent or dependent. In contrast, intracellular accumulation is a general term to describe the extent of an exogenous compound remains inside cells, which is an agglomerate effect determined by permeability, specific and non-specific interactions with intracellular proteins and other biomolecules, and elimination through metabolism and drug efflux pumps.

Although we have not completely understood why RC-1 has enhanced intracellular accumulation, which is out of the scope of this work, we suspect that RC-1 can be trapped inside cells through its fast and reversible reactions with the high concentration of glutathione (GSH) inside cells (usually between 1-10 mM). In contrast, RNC-1 cannot react with GSH. Technically, IRC-1 can irreversibly react with GSH but with a much slower reaction rate. This hypothesis could be tested by measuring the target engagement IC_{50} values of RC-1 at different intracellular GSH concentrations in HEK293 cells. If our hypothesis were correct, we would predict that RC-1 may show very low or undetectable concentrations on the receptor side in a CaCo-2 cell monolayer based permeability assay because RC-1 could be trapped inside CaCo-2 cells.

2. It will be useful to make another reversible noncovalent compound as reference, which is directly reduce the double bond of RC-1. Compare its cell permeability with RC-1 may give us more information, such as whether the enhanced permeability is caused by adding lipophilic groups (cyano and di-methyl) or because of the reversible covalent ability.

We thank the reviewer for this suggestion. We have made the two control compounds as shown below and compared their BTK target engagement with RC-1 (added to the revised manuscript as **Figure S6**). The BTK target engagement IC_{50} value of RC-1 is 10 and 28 folds of the values for IRC-1-DiMe and RNC-1-CN-DiMe, respectively. This indicates that the enhanced target engagement and intracellular accumulation is attributed to the cyano-acrylamide moiety instead of just the physical properties of the cyano and/or dimethyl groups.

3. Have the authors compared the BTK degradation between Crews's compound (such as MT802) and RC-1 in Momi-14 cell line or other cell lines? It will be useful to compare these two compounds, since MT802 is an optimal reversible noncovalent degrader.

We thank the reviewer for this suggestion. We compared the BTK degradation between MT802 and RC-1 in MOML-14 cells. The result showed that for BTK degradation in MOML-14 cells, RC-1 is more potent than MT-802 at a lower concentration (8 nM) and has a comparable potency at a high concentration (200 nM). This data has been added to the revised manuscript as SI, Figure S11.

4. In the *in vivo* PD study, the drug was dosed for 7 days. The PD effect is not as good what has been observed *in vitro*. Since degradation of BTK is already profound within 24 hours in cells, what was the reason for the 7-day dosing and for the relative less profound degradation of BTK?

This is also a great point that we did not discuss in detail. For the revised manuscript, we added the following:

Despite RC-1 is highly potent to degrade BTK in human cell culture and has a favorable PK property in mice, maintaining the plasma concentration above 200 nM over 24 h with a single *i.p.* injection of RC-1 (20 mg/kg), it is interesting to note that the maximum BTK degradation achieved in splenic B cells in mice is only ~50% even with 7 daily injections. To reconcile the discrepancy, we treated a mouse B cell lymphoma cell line derived from Eu-Myc mice with RC-1 *in vitro* and found that the maximum BTK degradation is only 30-40% even dosed up to 25 μ M of RC-1 (**Figure S10**), indicating that RC-1 is indeed much less potent for BTK degradation in mouse cells than in human cells. Although thalidomide and its derivatives, lenalidomide and pomalidomide, are clinically effective treatments for multiple myeloma, they are inactive in murine models *in vivo* due to the sequence difference between mouse and human Cereblon. Mice with a single I391V amino acid change in Crbn restore thalidomide-induced degradation of drug targets previously found in human cells. We rationalize that the BTK degradation potency difference for RC-1 in human and mouse cells may be due to the sequence variations for BTK and/or CRBN in different species, leading to inefficient formation of the ternary complex among RC-1, mouse Btk and mouse Crbn due to altered protein-protein interactions. This raised a question whether mice with humanized CRBN, either I391V and/or other mutations, should be developed to properly access the efficacy and toxicity of CRBN recruiting PROTACs in murine models, which can be a future research direction.

REVIEWERS' COMMENTS:

Reviewer #1 (Remarks to the Author):

The authors have adequately dealt with the previous comments that I raised.

Reviewer #2 (Remarks to the Author):

The authors have done a nice job to carefully address questions raised by all reviewers by performing additional experiments and making new compounds. I found the manuscript now acceptable for publication in Nature Communications.